# Decompile-Bench: Million-Scale Binary-Source Function Pairs for Real-World Binary Decompilation

**Hanzhuo Tan**[1,2]**, Xiaolong Tian**[1]**, Hanrui Qi**[1]**, Jiaming Liu**[1]**,**
**Zuchen Gao**[1,2]**, Siyi Wang**[1]**, Qi Luo**[1]**, Jing Li**[2]**, Yuqun Zhang**[1]*

[1]Research Institute of Trustworthy Autonomous Systems,
Southern University of Science and Technology
[2]Department of Computing, The Hong Kong Polytechnic University,
Research Centre for Data Science & Artificial Intelligence

## Abstract

Recent advances in LLM-based decompilers have been shown effective to convert low-level binaries into human-readable source code. However, there still lacks a comprehensive benchmark that provides large-scale binary-source function pairs, which is critical for advancing the LLM decompilation technology. Creating accurate binary-source mappings incurs severe issues caused by complex compilation settings and widespread function inlining that obscure the correspondence between binaries and their original source code. Previous efforts have either relied on used contest-style benchmarks, synthetic binary–source mappings that diverge significantly from the mappings in real world, or partially matched binaries with only code lines or variable names, compromising the effectiveness of analyzing the binary functionality. To alleviate these issues, we introduce Decompile-Bench, the first open-source dataset comprising two million binary-source function pairs condensed from 100 million collected function pairs, i.e., 450GB of binaries compiled from permissively licensed GitHub projects. For the evaluation purposes, we also developed a benchmark Decompile-Bench-Eval including manually crafted binaries from the well-established HumanEval and MBPP, alongside the compiled GitHub repositories released after 2025 to mitigate data leakage issues. We further explore commonly-used evaluation metrics to provide a thorough assessment of the studied LLM decompilers and find that fine-tuning with Decompile-Bench causes a 20% improvement over previous benchmarks in terms of the re-executability rate. Our code and data has been released in HuggingFace and Github. `https://github.com/albertan017/LLM4Decompile`

## 1 Introduction

Decompilation transforms the compiled binaries into high-level source code, helping engineers analyze program behaviors, uncover vulnerabilities, study malware, migrate legacy software, etc [11, 46, 83, 12, 30, 28, 31]. Decompilation is inherently difficult due to the information loss during the compilation process. Widely-adopted tools like Ghidra [32] and IDA [37] often fail to recover important details such as variable names [51, 84] and structural elements like loops and conditionals [76]. Motivated by the coding power of Large Language Models (LLMs) [71, 34, 68, 50, 64, 92, 75], researchers have developed LLM-based decompilers that demonstrate significant improvements in readability and accuracy of decompilation outputs [39, 7, 89, 40, 82, 58, 96, 59, 81, 74].

However, there lacks an open-source benchmark that provides large-scale binary-source function pairs drawn from real-world release-level binaries, which potentially hinders further advancements

---

*Yuqun Zhang is the corresponding author.

of LLM decompilation technologies. Specifically, while commercial efforts like BinaryAI [43], MLM [41], and ReCopilot [14] have assembled extensive private corpora, their closed-source nature complicates the access from academia. In contrast, a large-scale and publicly available benchmark would promote transparency, broaden participation, and accelerate progress in decompilation research. However, creating accurate binary-source mappings faces two major challenges. First, reconstructing the binary-source relationship often depends on DWARF debug information [18], which is rarely available in publicly released binaries. Second, aggressive compiler optimizations and pervasive function inlining [77] can obscure the correspondence between compiled binaries and their original source, rendering incomplete and unreliable mapping by directly tracking DWARF debug information. Previously, a group of researchers have used code contest benchmarks [76, 67] or developed tools to synthesize binary-source mappings [22, 6]. Although these benchmarks are somewhat beneficial for LLM-based decompilers, they differ significantly from real-world scenarios, raising concerns on their practical applicability. Alternatively, other researchers focus on linking binaries with fragments of source code. They tend to either target variable names and types [51, 84, 16], establishing connections between binary functions and their corresponding named entities in the source code, or directly extract line-level mappings from the DWARF debug information [57]. While these approaches provide valuable partial information, they neither resolve the fundamental obscured mapping problem, nor offer a complete understanding of binary functionality.

In this paper, we propose the first million-scale binary-source benchmark for real-world decompilation, namely Decompile-Bench, a corpus of two million function-level binary–source pairs condensed from 100 million collected pairs, i.e., 450 GB of binaries built from permissively licensed GitHub projects [63]. We also propose a pipeline Compile-Trace-Filter framework (CTF framework), to automate project builds, accurately trace each binary function back to its corresponding source code, and rigorously filter out low-quality data. We then augment Decompile-Bench with Decompile-Bench-Eval for the evaluation purposes with (1) manually crafted binaries drawn from two widely-used code-completion benchmarks HumanEval [15] and MBPP [8] and (2) binaries compiled from GitHub repositories published after 2025 for preventing potential data leakage issues.

Finally, we include three metrics commonly employed by recent LLM-based decompilers and benchmark them on Decompile-Bench-Eval. We also compare previous decompilers against those trained on Decompile-Bench, and find that Decompile-Bench collects high-quality binary-source function pairs, e.g., having a 20% improvement on re-executability over previous benchmarks. Our code and data has been released in HuggingFace[2] and Github. Our main contributions can be summarized as follows.

- Decompile-Bench. We propose the first million-scale benchmark for real-world decompilation with two million binary-source function pairs. Our pipeline ensures precise mappings and filters 100 million raw function pairs down to two million high-quality ones.

- Decompile-Bench-Eval. We propose a leakage-resistant evaluation suite containing (1) hand-crafted binaries from HumanEval and MBPP and (2) binaries built from GitHub repositories published after 2025.

- Comprehensive evaluation. Our experiments demonstrate that fine-tuning with our benchmark improves re-executability by over 20% compared to the previous benchmarks.

## 2  Related Work

Despite extensive work on binary-only benchmarks, e.g., compiler provenance [26], vulnerability detection [52, 9, 65, 38], summarization [45, 88] and similarity search [80, 47, 72, 55], pairing binaries with their original source code has received less attention.

Commercial efforts like BinaryAI [43], MLM [41], and ReCopilot [14] have collected large-scale private binary–source corpora, yet their closed-source nature limit access from academia. For open-source binary–source benchmarks, we categorize them into four groups (Figure1) as follows. Note that since disassembling a binary to assembly is rather deterministic, we treat "binary" and "assembly" as equivalent terms in this paper.

**Code Contest Benchmarks.** Built from programming-contest problems, these benchmarks [76, 67] may include unit tests that verify whether decompiled output actually executes, and they typically

---

[2]https://huggingface.co/datasets/LLM4Binary/decompile-bench

| Category | Code Contest | Synthetic |
|---|---|---|
| Benchmarks | Decompile-Dataset [26] HumanEval-Decompile [11] | AnghaBench [27] ExeBench [28] CSMith [48] |
| Samples | int func0(float num [], int size, float threshold){ int i, j for (i = 0; i < size; i++) for (…) if (…)…} **a** | struct of_device{int dummy; } ; struct device {int dummy; } ; struct device* bus_find_device (int /*<<< orphan*/ *,..) ; int /*<<< orphan*/ …;} **b** |

| Category | Fragment-Level | Real-World Function-Level |
|---|---|---|
| Benchmarks | Dire[9] Dirty [29] Resym [10] Assemblage [30] | CodeCMR [53] Idioms [54] **Decompile-Bench** |
| Samples | source code / rva / func
1 return J();... 0x00004320 46
2 return t->size();... 0x00001A80 55
3 column++;... 0x000034F0 58
4 if (lookahead &... 0x0000B050 **c** | Game::Game(std::size_t grid_width, std::size_t grid_height) : snake(grid_width, grid_height), snake2(grid_width, grid_height), engine(dev()), … { snake.setPosition(…)…} **d** |

Figure 1: Different types of binary–source benchmarks.

involve functions with simple variable names and standard types such as "int" or "float" (Figure1-a), omitting user-defined types that are common in real-world scenarios.

**Synthetic Mapping Benchmarks.** Built on synthetic data, researchers [22, 6] develop tools to generate synthetic types and dependencies, e.g., `struct of_device {int dummy}`, `int /*<<< orphan*/` as in Figure1-b, to enable compilation. While these benchmarks have motivated multiple LLM-driven decompilers [76, 7, 89, 44], their synthetic nature also raises issues on decompiling real-world binaries. More details on these datasets are discussed in Appendix A. Other works [13, 61] use Csmith [87] to randomly generate compilable C programs. Both approaches do not represent real-data distributions.

**Fragment-Level Benchmarks.** These benchmarks focus on linking binaries with fragments of source code including token-level mapping and line-level mapping. Specifically, token-level mapping indicates that name/type recovery [51, 84, 16, 19, 69, 23, 54] only reconnects identifiers with the source. It does not attempt to reconstruct control-flow structures, data-flow relationships, or high-level abstractions. As a result, it fails to produce a standalone, compilable program, limiting its effectiveness for downstream binary analysis tasks. A typical line-level mapping benchmark Assemblage [57] includes detailed metadata and tools for a variety of binary-centric analyses. Although its primary emphasis is on the binary side, it also offers partial binary-source mapping by tracing DWARF debug information to link binary instructions back to their original source code lines. However, compiler optimizations severely disrupt this process such that the line-to-line correspondences are largely lost. As shown in Figure1-c, while one can still map a source code line to its binary via the Relative Virtual Address (RVA), those RVAs may span different functions. Consequently, the original code context around each line is lost and the function body becomes highly fragmented. While these benchmarks provide useful partial insights, they do not deliver a full representation of binary functionality.

**Real-World Function-Level Benchmarks.** In these benchmarks, the complete function-level source code including full function signatures, bodies, user-defined types, and names is extracted from real-world GitHub repositories. Accordingly, binaries are built with standard toolchains and linked against external libraries, closely replicating publicly released executable versions to form the real-world binary-source function pairs. However, the existing real-world function-level benchmarks such as CodeCMR [90] (with 60K pairs) and Idioms [25] (with 154K pairs) are both orders of magnitude small for modern LLM training.

To conclude, there lacks a real-world function-level and publicly accessible large-scale benchmark to promote transparency, attract more contributors, and foster progress in the decompilation research.

## 3 Decompile-Bench

In this section, we first introduce the CTF (**C**ompile-**T**race-**F**ilter) framework, the pipeline used to build our benchmark, and then provide statistics and analysis of Decompile-Bench.

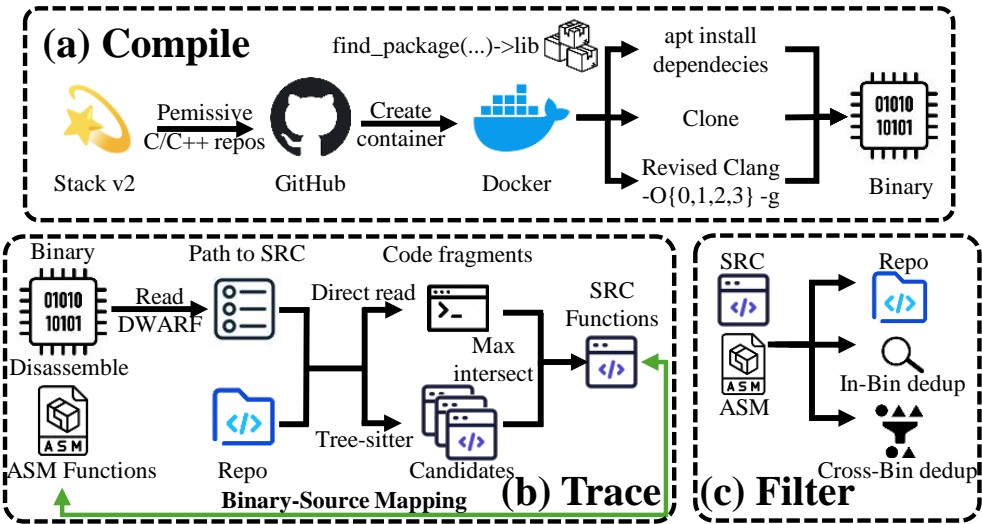

Figure 2: The CTF framework which (a) automates compilation, (b) precisely traces binary–source mappings, and (c) rigorously filters out low-quality data.

### 3.1 CTF framework

Figure 2 illustrates the CTF framework we use to build Decompile-Bench. As mentioned, there are two issues to link binaries back to their sources: (1) DWARF debug information is typically stripped from production builds, and (2) heavy compiler optimizations and function inlining radically transform code structure. They together make DWARF-only alignment incomplete and error-prone. To address both issues, our CTF framework automates project compilation, precisely traces function-level binary-source mappings, and applies robust filters to retain only high-quality pairs.

**Automatic Compilation.** Prior works (e.g., Dirty [16] and Idioms [25]) use tools like GHCC [42] to inject optimization and debug flags by setting environment variables (e.g., `env["MOCK_GCC_OVERRIDE_FLAGS"]`, `env["CMAKE_CXX_FLAGS"]`), expecting that the compiler would recognize these settings. However, in reality, a substantial proportion of projects ignore these environment flags. Additionally, compiling C/C++ projects is challenging due to numerous dependencies and libraries not included in the projects, often requiring manual intervention. As a result, direct compilation typically fails for most projects.

We fork Clang [77] to address the aforementioned issue. In particular, we patch its driver and invocation logic to force our desired `-O{0,1,2,3}` and `-g` flags, then rebuild Clang and symlink all compiler calls (e.g. `env["CC"]`, `/usr/bin/clang`) to our patched compiler. For the missing packages, we automatically parse CMakeLists.txt of each project for `find_package(...)` directives, query GPT for the correct install commands, and cache a mapping from package name to install recipe. During the build, we pre-install any missing libraries before invoking CMake. This end-to-end toolchain guarantees consistent optimization/debug settings and includes DWARF debug information, improving the overall compilation success rate. This toolchain also supports Makefiles, as detailed in Appendix F.

**Trace Binary-Source Correspondence.** DWARF debug data records file paths and line numbers that link line-level assembly back to source. In practice, however, compiler optimizations and function inlining make these mappings incomplete, fragmented, or out of order. To solve this, we introduce the Source-Trace algorithm for accurate function-level binary-source pairing. As shown in Algorithm 1, our Source-Trace algorithm takes as input a binary $B$ and its originating source project $S$, and produces a mapping $M$ from each binary function $f_b \in B$ to the corresponding complete source function $f_s \in S$.

For each binary function $f_b$ (Line 1), we extract all of its DWARF-reported source locations and collect them into a set `func_segment` (Line 2). This replicates the "conventional" binary-source mapping step as employed in previous works [89, 57], which yields only fragmented, unordered

---

**Algorithm 1** Pairing Binary Functions with Source Functions

---

**Input:** Binary project $B$, source project $S$
**Output:** Map $M$, containing the mapping between each binary function in $B$ to a source function
 1: **for all** binary function $f_b$ in $B$ **do**
 2:    Extract DWARF information in $f_b$ to obtain corresponding source-code lines; group them as *func_segment*
 3:    Candidates $\leftarrow \emptyset$
 4:    **for all** source line $\ell$ in *func_segment* **do**
 5:       Use Tree-sitter on $S$ to extract the complete source function $f_s$ containing $\ell$
 6:       Candidates $\leftarrow$ Candidates $\cup \{f_s\}$
 7:    **end for**
 8:

$$f_s^* \leftarrow \arg \max_{f_s \in \text{Candidates}} \left| \textit{func\_segment} \cap \text{Lines}(f_s) \right|$$

 9:    $M[f_b] \leftarrow f_s^*$
10: **end for**
11: **return** $M$

---

snippets. For each line $l$ in `func_segment` (Lines 3–7), we use Tree-sitter [20] to parse the project $S$, locate the full source function surrounding that line, and add it to a candidate list. After parsing, each candidate is a complete function (signature + body). We then compute the line intersection between `func_segment` and each candidate function and choose the candidate with the largest overlap as the true source match for $f_b$ (Line 8). At last, we record the mapping between $f_b$ and $f_s$ and return the final mapping $M$ for all functions in $B$ (Lines 9-11).

This procedure fixes missing or reordered source code fragments by realigning them with clear function boundaries, yielding precise binary–source function pairs. For validating the matching algorithm, please refer to Appendix E.

**Filter data.** The raw mappings produced by our Source-Trace algorithm still contain a large amount of noise, such as trivial system header functions, multiple binary functions pointing to the same overloaded or template source function, and duplicate source functions. To enhance the quality of Decompile-Bench, we therefore apply a three-stage filtering pipeline as follows.

(1) Project-scope filter. We remove any source function not actually defined in the target repository. These are mostly trivial helpers, e.g., get() and set(), extracted from system or dependency headers, which offer less value for decompilation. (2) In-binary deduplicator. Whenever several functions within one binary claim the same source function (usually due to template instantiation), we keep only the single best match, i.e., the one with the largest DWARF-segment intersection as computed in Algorithm 1, Line 8, and discard the rest. (3) Cross-binary deduplicator. We apply MinHash-LSH [10] to the remaining source functions and assemblies to eliminate near-duplicate, following standard corpus-cleaning practices in LLM training [71, 63, 48, 5].

## 3.2 Data Statistics and Analysis

To ensure permissive licensing, we select the **C/C++** GitHub repositories from the Stack V2 [63], whose licenses are either approved by the Blue Oak Council [1] or flagged "Permissive" by Scan-Code [2]. We further require at least one GitHub star for basic quality assurance [5] and the presence of a `CMakeLists.txt` to streamline builds. Each project is then compiled at four optimization levels (`-O{0,1,2,3}`). In total, we successfully built 3,961 repositories, yielding roughly 450 GB of executables across 85K binaries and extracting about 100 million binary functions. This number far exceeds the roughly 5 million source functions present across the projects, highlighting the need for data filtering to improve the quality. The filtering statistics and examples are summarized in Figure 3. In particular, Figure 3-a shows that 45% of binary functions are the system or dependency headers associated with our collected repositories. Removing duplicates within each binary eliminates about 20% of data, and 32% of the binary functions are removed via cross-binary deduplicator (please refer to Appendix B for detailed analysis). Finally, our compact benchmark retains only 2% of the raw data (two million functions out of 100M). As shown on the Figure 3-b, removing codes from header files or duplicates eliminates the vast majority (40% of all raw data) of short snippets (under five lines)

that offer limited training information for an LLM decompiler. Figure 3-c shows representative cases removed by the project-scope filter (upper subfigure) and the in-binary deduplicator (lower subfigure). The project-scope filter mostly discards trivial helpers (e.g., get()), constructors, and destructors from system or dependency headers. In-binary duplicates stem from template instantiation. In particular, when a template is instantiated across multiple object files, the compiler emits separate, mangled versions of identical logic (e.g., the code snippet of the lower subfigure of Figure 3-c is compiled to 17 binary functions within the same binary). Although generated independently from the same source, these duplicates offer no additional diversity. The ablation study of Section 5.3.2 further affirms the effectiveness of this filtering step, which is consistent with prior studies [63, 48, 5], i.e., eliminating duplicates markedly improves performance across all experiments.

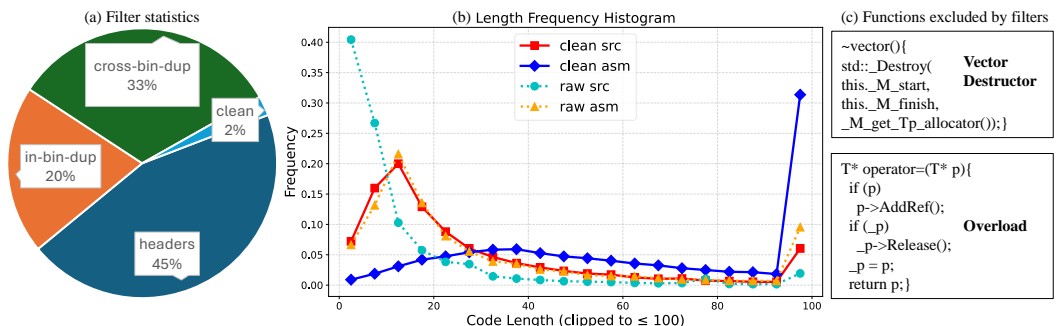

Figure 3: (a) Filter statistics. (b) Length-frequency histogram. "Clean src" denotes filtered code, while "raw asm" denotes unfiltered output. (c) Sample functions excluded by the filters.

**Ethics.** Our Decompile-Bench is composed solely of publicly available code from permissively licensed GitHub repositories [63] (e.g. MIT, BSD, and Apache 2.0), which precludes large amount of non-permissive libraries. Commercial software remains protected through obfuscation techniques that render effective decompilation impossible, as demonstrated in previous research [76]. Additionally, the benchmark serves for purely academic and educational purposes, supporting legitimate research in binary analysis and reverse engineering education.

## 4 Decompile-Bench-Eval

In this section, we discuss the construction of Decompile-Bench-Eval with the commonly-used evaluation metrics in decompilation.

### 4.1 Dataset

We first include two widely-used code-completion benchmarks, HumanEval [15] and MBPP [8], which by convention must be held out of the training data of any LLM. Specifically, we translate each Python solution and its test cases of the HumanEval and MBPP into **C/C++** manually [76, 94]. Note that unlike HumanEval-Decompile [76], which is restricted to C, our dataset offers language support for both C and C++. At test time, we compile each sample into a binary at four optimization levels (-O0—O3), decompile it back to source, and run the original tests to ensure correctness.

To augment the real-world applicability of Decompile-Bench-Eval, we also collect 121 GitHub repositories created after 2025 with more than one star as in the Decompile-Bench. We ensure permissive licenses, and then use the CTF framework to compile and clean each project. To eliminate any overlap with known training corpora, we strip out "third-party", "external", and the submodule directories. In total, this yields about 60K C/C++ new functions to alleviate the data leakage concerns. We name this dataset as GitHub2025 in this paper. In addition, we include ProRec [73] for evaluation.

## 4.2 Metrics

We evaluate decompilers using three core metrics: functionality recovery (Re-Executability [7]), readability (R2I [27]), and text similarity (edit similarity [53]). We also include additional evaluation metrics, i.e., embedding-based similarity [93] and CodeBLEU [70], in Appendix C.

**Re-Executability.** Re-executability measures whether the decompiled function's functionality matches that of the original source function, which is widely adopted in decompilation [76, 7, 89, 44, 29]. For source function $s$ and decompiled function $d$, the decompiled function $d$ is re-executable if they produce identical outputs on all inputs. Formally,

$$\text{ReExe}_s(d) \iff \forall x \in X, s(x) = d(x) \tag{1}$$

In practice, we approximate the solution of $\text{ReExe}_s(d)$ over a finite test set. Given the unit tests $T$, we define the Re-Executability for $d$ as: $\text{ReExe}_s(d)$ if $\forall x \in T, \ s(x) = d(x)$. Re-executability is also called the I/O accuracy [7] or pass rate [44].

**R2I.** Relative Readability Index (R2I) [27] is a metric devised for the relative, quantitative evaluation of decompiled C code. Given a set of decompiler outputs, R2I produces a normalized score between 0 and 1. To compute R2I, we typically construct an abstract syntax tree (AST) [3] first and then extract pre-defined features from the AST. At last, we calculate R2I with the feature weights.

**Edit Similarity.** Based on Levenshtein Distance [53], this metric captures the minimum number of insertions, deletions, or substitutions needed to turn the generated code into the reference. It is a popular measure in decompilation to assess readability [76, 7].

# 5 Experiments

## 5.1 Setups

We compiled our binaries on Ubuntu 20.04 targeting the Linux x64 architecture, using Clang-19 with the C++17 standard. For fine-tuning, we initialized our model with the 1.3B and 6.7B LLM4Decompile-End checkpoint [76]—a state-of-the-art LLM-based decompiler pre-trained on 15 billion tokens from ExeBench [6]. We adopt the sequence-to-sequence (S2S) [21, 79] prediction as our training objective, following the practice in LLM-decompilers [76, 7, 44]. As a preliminary study, we use 10% of the Decompile-Bench function pairs as the training data and train the studied models using LLaMA-Factory library [95] with a $batch\ size = 64$ and $learning\ rate = 5e{-}6$ for one epoch (0.2B tokens). We name the resulting model as LLM4Decompile-DCBench. All the experiments are performed on NVIDIA A800-80GB GPU clusters. Fine-tuning the 1.3B model takes 4 hours on $8 \times A800$, while the 6.7B model takes 1 day. For evaluation, we use the *vllm* [49] to accelerate the generation (decompilation) process, with the max number of new generated tokens set to 512. We employ greedy decoding to minimize randomness.

## 5.2 Baselines

We adopt IDA [37]—the most widely used traditional decompiler—as our primary baseline. We also evaluate state-of-the-art commercial models GPT-4.1-mini [68] and Claude-Sonnet-4-reasoning. LLM4Decompile-End [76] and Idioms [25] are included as our open-source baselines. Other LLM-based decompilers like Nova [44], Ref-Decomp [29] do not illustrate their preprocessing approaches, hindering our replications. Thus, they are not included for evaluations.

## 5.3 Results

### 5.3.1 Main Results

Table 1 reports the re-executability rate across optimization levels (O0–O3) on HumanEval and MBPP. GPT-4.1-mini, although not explicitly trained for decompilation, demonstrates robust decompilation performance, achieving average re-executability rates of 13.42% and 19.89% across the two benchmarks. By fine-tuning the LLM4Decompile-End-1.3b on just 10% of our Decompile-Bench data (i.e., LLM4Decompile-DCBench-1.3b), we improve re-executability of LLM4Decompile-End

Table 1: Comparisons of different decompilers in terms of re-executability

| Re-Executability Rates | HumanEval | | | | | MBPP | | | | |
|---|---|---|---|---|---|---|---|---|---|---|
| | O0 | O1 | O2 | O3 | AVG | O0 | O1 | O2 | O3 | AVG |
| GPT-4.1-mini | 21.95 | 11.58 | 10.07 | 10.06 | 13.42 | 31.37 | 16.74 | 16.64 | 14.79 | 19.89 |
| IDA | 18.60 | 19.81 | 17.69 | 16.77 | 18.22 | 25.62 | 25.05 | 23.72 | 23.57 | 24.49 |
| Idioms-1.3b | 30.56 | 16.10 | 12.63 | 12.36 | 17.91 | 33.97 | 20.47 | 18.13 | 17.30 | 22.47 |
| LLM4Decompile-End-1.3b | 26.22 | 12.81 | 14.03 | 13.42 | 16.22 | 29.16 | 16.99 | 17.92 | 18.07 | 20.54 |
| LLM4Decompile-DCBench-1.3b | 33.23 | 18.60 | 16.47 | 15.24 | 20.89 | 35.06 | 21.56 | 22.80 | 20.28 | 24.93 |
| LLM4Decompile-DCBench-6.7b | 61.59 | 30.18 | 34.15 | 32.01 | 39.48 | 58.32 | 39.58 | 39.73 | 37.06 | 43.67 |
| Claude-Sonnet-4-reasoning | 65.85 | 42.68 | 39.63 | 39.02 | 46.79 | 67.76 | 51.69 | 53.02 | 50.25 | 55.68 |

Table 2: Main comparison of different decompilers for R2I on evaluation benchmarks.

| R2I | HumanEval | | | | | MBPP | | | | | GitHub2025 | | | | | ProRec Score |
|---|---|---|---|---|---|---|---|---|---|---|---|---|---|---|---|---|
| | O0 | O1 | O2 | O3 | AVG | O0 | O1 | O2 | O3 | AVG | O0 | O1 | O2 | O3 | AVG | |
| GPT-4-1-mini | 62.38 | 52.63 | 55.68 | 53.90 | 56.14 | 61.79 | 55.34 | 57.05 | 55.83 | 57.50 | 51.65 | 39.64 | 46.62 | 55.83 | 48.43 | 55.01 |
| IDA | 41.49 | 36.29 | 35.85 | 35.32 | 37.23 | 41.82 | 34.87 | 35.16 | 36.21 | 37.02 | 45.87 | 38.85 | 36.99 | 36.20 | 39.48 | 38.35 |
| Idioms-1.3b | 68.18 | 66.92 | 67.46 | 65.48 | 67.01 | 69.12 | 67.01 | 63.91 | 62.35 | 65.60 | 63.26 | 53.26 | 51.19 | 56.07 | | 64.86 |
| LLM4Decompile-End-1.3b | 65.69 | 60.48 | 60.66 | 59.37 | 61.55 | 67.93 | 63.47 | 65.69 | 63.01 | 65.03 | 54.26 | 51.73 | 53.42 | 50.56 | 52.49 | 57.49 |
| LLM4Decompile-DCBench-1.3b | 68.93 | 68.74 | 69.03 | 67.76 | 68.62 | 69.13 | 70.97 | 68.03 | 67.79 | 68.98 | 64.40 | 65.72 | 61.74 | 63.31 | 63.79 | 65.73 |
| LLM4Decompile-DCBench-6.7b | 69.35 | 68.91 | 69.79 | 68.42 | 69.12 | 72.30 | 71.99 | 72.25 | 70.67 | 71.80 | 72.67 | 70.23 | 66.55 | 67.76 | 69.30 | 66.15 |
| Claude-Sonnet-4-reasoning | 61.09 | 54.94 | 55.65 | 55.28 | 56.74 | 64.78 | 60.62 | 61.53 | 61.71 | 62.16 | 55.70 | 43.88 | 45.04 | 51.71 | 49.08 | 57.38 |

Table 3: Main comparison of decompilers for Edit Similarity on evaluation benchmarks.

| Edit Similarity | HumanEval | | | | | MBPP | | | | | GitHub2025 | | | | | ProRec Score |
|---|---|---|---|---|---|---|---|---|---|---|---|---|---|---|---|---|
| | O0 | O1 | O2 | O3 | AVG | O0 | O1 | O2 | O3 | AVG | O0 | O1 | O2 | O3 | AVG | |
| GPT-4-1-mini | 46.09 | 33.83 | 34.75 | 29.66 | 36.08 | 47.52 | 37.34 | 39.15 | 32.63 | 39.16 | 21.15 | 18.64 | 19.38 | 18.43 | 19.40 | 34.74 |
| IDA | 25.47 | 21.01 | 20.18 | 17.92 | 21.15 | 27.66 | 23.63 | 22.01 | 19.43 | 23.18 | 22.17 | 18.21 | 19.69 | 18.93 | 19.75 | 27.24 |
| Idioms-1.3b | 48.84 | 38.08 | 35.93 | 34.66 | 39.35 | 49.35 | 38.13 | 35.91 | 34.36 | 39.44 | 30.27 | 24.04 | 25.09 | 24.18 | 25.90 | 36.03 |
| LLM4Decompile-End-1.3b | 43.37 | 36.91 | 36.76 | 36.30 | 38.34 | 44.82 | 39.67 | 39.01 | 38.13 | 40.41 | 23.09 | 20.61 | 21.77 | 20.81 | 21.57 | 34.26 |
| LLM4Decompile-DCBench-1.3b | 54.36 | 43.54 | 44.21 | 42.78 | 46.22 | 56.38 | 48.14 | 46.76 | 45.79 | 49.28 | 30.99 | 29.21 | 30.23 | 27.59 | 29.51 | 38.85 |
| LLM4Decompile-DCBench-6.7b | 62.32 | 51.91 | 51.66 | 52.99 | 54.72 | 64.12 | 55.40 | 53.87 | 53.39 | 56.70 | 34.29 | 32.74 | 34.18 | 29.96 | 32.79 | 45.23 |
| Claude-Sonnet-4-reasoning | 60.75 | 48.45 | 47.12 | 46.40 | 50.68 | 64.64 | 54.27 | 53.10 | 51.98 | 55.99 | 36.29 | 32.80 | 33.12 | 31.32 | 33.38 | 41.99 |

to 20.89% on HumanEval (a 28.8% relative gain) and 24.93% on MBPP (a 21.4% gain) compared with the baseline model, demonstrating the advantage of adopting the benchmark with real-world binary-source function pairs.

The recent model Claude-Sonnet-4 achieves a high re-executability rate on HumanEval (46.79%), demonstrating the power of large-scale commercial LLMs. Idioms shows promise on O0 optimizations (30.56% re-executability) but its performance degrades sharply on higher optimizations like O2/O3 (~12%). This suggests its training data may lack sufficient examples of aggressively optimized code, validating the effectiveness of our Automatic Compilation technique (Section 3.1) which ensures that such cases are well represented. The LLM4Decompile-DCBench-6.7b model achieves a 39.48% re-executability rate, which is highly competitive with Claude's 46.79%. It indicates that a model trained on a small fraction (10%) of our high-quality data can approach the performance of a massive commercial LLM trained on trillions of tokens.

Table2 shows the R2I performance for the studied decompilers. Note that R2I applies only to C programs, so we report its scores exclusively on C functions. Although IDA deliver comparable re-executability rates with LLM-based decompilers, its R2I readability index is only about 40, i.e., nearly half the 70 R2I achieved by LLM-based decompilers. The low R2I scores of conventional decompilers highlight their poor readability and underscore the need for LLM-based decompilers to produce more human-readable output for facilitating binary analysis.

Moreover, on the real-world decompilation test set GitHub2025, LLM4Decompile-DCBench-1.3b trained on real-world binary–source function pairs delivers a 21.5% higher R2I score than LLM4Decompile-End-1.3b, which was fine-tuned on synthetic data. This result highlights the importance of real-world training data for producing executable and readable decompiled code. We also observe the similar gains in terms of edit similarity (Table 3), where LLM4Decompile-DCBench-1.3b achieves a 36.8% improvement over the baseline LLM4Decompile-End-1.3b.

We also evaluate additional metrics, i.e., embedding-based similarity, CodeBLEU, and GPT evaluation, and observe consistent trends as in Tables 1, 2, and 3 Please refer to Appendix C for details.

### 5.3.2 Ablations

Following the setup in Section 5.1, we also perform a series of ablation studies by comparing Decompile-Bench with other benchmarks via fine-tuning LLM4Decompile-End on an equal amount of 200K binary–source function pairs drawn from two other sources, i.e., the 100M-pair raw corpus of Decompile-Bench (Section 3.2) and ExeBench.

Table 4: Ablation study on training data for re-executability rate. The tag "+ExeBench" denotes that the data come from ExeBench, whereas "+Decompile-Bench-raw" indicates the use of the unfiltered Decompile-Bench as discussed in Section 3.2.

| Re-Executability Rates | HumanEval | | | | | MBPP | | | | |
|---|---|---|---|---|---|---|---|---|---|---|
| | O0 | O1 | O2 | O3 | AVG | O0 | O1 | O2 | O3 | AVG |
| LLM4Decompile-End | 26.22 | 12.81 | 14.03 | 13.42 | 16.22 | 29.16 | 16.99 | 17.92 | 18.07 | 20.54 |
| +Exebench | 26.22 | 13.89 | 13.11 | 13.89 | 16.78 | 27.16 | 17.66 | 18.74 | 17.25 | 20.20 |
| +Decompile-Bench-raw | 24.70 | 13.41 | 13.11 | 12.20 | 15.86 | 26.49 | 16.48 | 15.40 | 14.32 | 18.17 |
| +Decompile-Bench | 33.23 | 18.60 | 16.47 | 15.24 | 20.89 | 35.06 | 21.56 | 22.80 | 20.28 | 24.93 |

Table 5: Ablation study on training data for R2I metric. Note that since R2I evaluates decompiled code in a relative context quantitatively [27], its values can vary significantly for the same decompiler when compared with different baselines.

| R2I | HumanEval | | | | | MBPP | | | | | GitHub2025 | | | | |
|---|---|---|---|---|---|---|---|---|---|---|---|---|---|---|---|
| | O0 | O1 | O2 | O3 | AVG | O0 | O1 | O2 | O3 | AVG | O0 | O1 | O2 | O3 | AVG |
| llm4decompile | 51.01 | 55.54 | 57.90 | 53.83 | 54.57 | 54.14 | 55.10 | 57.61 | 56.15 | 55.75 | 47.21 | 45.15 | 51.78 | 50.68 | 48.71 |
| +Exebench | 50.74 | 56.77 | 56.62 | 54.47 | 54.40 | 55.67 | 55.28 | 57.09 | 55.39 | 56.11 | 48.47 | 46.35 | 48.42 | 48.65 | 47.97 |
| +Decompile-Bench-raw | 52.82 | 57.64 | 55.55 | 55.31 | 55.33 | 56.00 | 57.75 | 56.71 | 56.87 | 56.83 | 70.02 | 62.09 | 67.79 | 64.48 | 66.10 |
| +Decompile-Bench | 55.59 | 59.87 | 62.90 | 63.06 | 60.36 | 60.06 | 62.01 | 63.54 | 61.64 | 61.81 | 73.34 | 69.72 | 70.79 | 69.73 | 70.89 |

Table 6: Ablation study on training data for edit similarity.

| Edit Similarity | HumanEval | | | | | MBPP | | | | | GitHub2025 | | | | |
|---|---|---|---|---|---|---|---|---|---|---|---|---|---|---|---|
| | O0 | O1 | O2 | O3 | AVG | O0 | O1 | O2 | O3 | AVG | O0 | O1 | O2 | O3 | AVG |
| llm4decompile | 43.37 | 36.91 | 36.76 | 36.30 | 38.34 | 44.82 | 39.67 | 39.01 | 38.13 | 40.41 | 23.09 | 20.61 | 21.77 | 20.81 | 21.57 |
| +Exebench | 43.36 | 35.30 | 34.99 | 34.74 | 37.10 | 45.46 | 39.91 | 38.99 | 38.43 | 40.69 | 22.70 | 20.30 | 21.43 | 20.52 | 21.24 |
| +Decompile-Bench-raw | 48.35 | 37.39 | 36.99 | 36.56 | 39.82 | 49.38 | 40.45 | 39.67 | 38.81 | 42.08 | 30.19 | 28.66 | 29.69 | 27.50 | 29.01 |
| +Decompile-Bench | 54.36 | 43.54 | 44.21 | 42.78 | 46.22 | 56.38 | 48.14 | 46.76 | 45.79 | 49.28 | 30.99 | 29.21 | 30.23 | 27.59 | 29.51 |

Table 4 summarizes the re-executability rate of LLM-decompilers trained on different benchmarks. Since the base model LLM4Decompile-End is trained on ExeBench and additional training from the same source does not provide further benefits, "LLM4Decompile-End+Exebench" model achieves similar results to the base model LLM4Decompile-End. For the model trained on Decompile-Bench-raw, although the data is sampled from real world, the low-quality nature leads to 2.2% and 11.4% re-executability decline against to the base model. Compared to the Decompile-Bench-raw data, fine-tuning with the clean and compact data from Decompile-Bench significantly improves the re-executablility of the base model for over 20% as discussed in the main results.

Tables 5 and 6 report R2I and edit-similarity scores for LLM-based decompilers trained on different benchmarks. The model using Decompile-Bench-raw outperforms ExeBench by about 2% on both metrics for HumanEval and MBPP. However, the re-executability results in Table 4 indicate that these small readability gains come at the cost of increased logical errors, namely hallucination [66, 56]i.e., the model produces plausible but incorrect results. By training on our Decompile-Bench, we achieve around averaged 15% improvement in readability and similarity compared to LLM4Decompile-End with markedly higher correctness.

### 5.4 Extending Decompile-Bench to Search and Other Tasks

The binary–source pairs in Decompile-Bench also directly support a binary–source search task. In our preliminary experiments, an embedding model trained on just 10% of Decompile-Bench data achieves a 27% recall@1 on Decompile-Bench-Eval where detailed results are provided in Appendix D.

In addition to binary-source search, as Decompile-Bench includes binaries from the same projects built under different optimization settings, it naturally supports training binary-similarity-oriented

techniques [72, 62]. Moreover, since each binary is paired with its complete source function (signature and body) in Decompile-Bench, variable names and types can be extracted to facilitate type-recovery tasks [85, 73].

### 5.4.1 Case Study

<div>

**(a) Source code**

```
my_bool bitmap_is_set_all(const MY_BITMAP *map){
  my_bitmap_map *data_ptr= map->bitmap;
  my_bitmap_map *end= map->last_word_ptr;
  for (; data_ptr < end; data_ptr++)
    if (*data_ptr != 0xFFFFFFFF)
      return FALSE;
  return (*data_ptr|map->last_word_mask)==0xFFFFFFFF;}
```

**(d) LLM4Decompile-End**

```
bool bitmap_is_set_all(const struct bitmap *b){
  const struct bitmap_header *h = (const struct
bitmap_header *)b;
  const unsigned int *p = (const unsigned int *)h;
  const unsigned int *end = (const unsigned int *)h->end;
  bool ret;
  if (p < end) {
    if (*p != 0xffffffff) {ret = false;}
    else {p++;
      while (p < end) {
        ret = (*p++ | h->mask) == 0xffffffff;}}}
  else {ret = true;}
  return ret;}
```

**(b) IDA Pro**

```
bool  bitmap_is_set_all(long long a1){
    _DWORD *i; // [rsp+8h] [rbp-18h]
    for ( i = *(_DWORD **)a1; (unsigned long long)i
< *(_QWORD *)(a1 + 8); ++i ){
      if ( *i != -1 ) return 0;}
    return (*(_DWORD *)(a1 + 24) | *i) == -1;}
```

**(c) GPT4.1-mini**

```
bool bitmap_is_set_all(bitmap_t* bitmap) {
  uint64_t* current = bitmap->addr;
  uint64_t size_in_words=bitmap->size/sizeof(uint64_t);
  for (uint64_t i = 0; i < size_in_words; ++i) {
    if (current[i] != UINT64_MAX) {return false;}}
  return true;}
```

**(f) LLM4Decompile-DCBench**

```
bool bitmap_is_set_all(const struct bitmap *bitmap){
  const uint8_t *p = bitmap->data;
  const uint8_t *end = bitmap->end;
  while (p < end) {
    if (*p != UINT8_MAX) return false;
    p++;}
  return (*p | bitmap->mask) == UINT8_MAX;}
```

</div>

Figure 4: A case study on a bitmap function with (a) source code and the decompilation results from (b) IDA Pro, (c) GPT4.1-mini, (d) LLM4Decompile-End, and (e) LLM4Decompile-DCBench

Figure 4 presents a case study on a bitmap function and its corresponding decompilation results across different tools. IDA Pro preserves the control flow but produces unreadable output, with complex type conversions involving pointer arithmetic (e.g., `(_DWORD *)(a1 + 24)))`) and generic variable names. GPT-4.1-mini, on the other hand, excels at generating human-readable names and types (e.g., `bitmap_t bitmap` close to the ground truth `MY_BITMAP map`) but often compromises logical correctness, such as hallucinating a `return true;`, while ignoring the critical return comparison logic. LLM4Decompile-End and LLM4Decompile-DCBench could recover correct control flow and logic while producing meaningful identifiers.For example, LLM4Decompile-DCBench names a variable `const uint8_t *end = bitmap->end`, which accurately reflects its functionality.

## 6 Conclusion

In this paper, we present Decompile-Bench, the first large-scale decompilation benchmark containing two million function-level binary–source pairs, sourced from 100 million pairs in permissively licensed GitHub projects. In particular, we develop the CTF framework, an automated pipeline that compiles projects, traces each binary function to its source code, and rigorously filters out low-quality examples. For evaluation purposes, we extend Decompile-Bench with Decompile-Bench-Eval, which comprises (1) manually crafted binaries from the HumanEval and MBPP code-completion challenges and (2) binaries compiled from GitHub repositories published after 2025 to prevent data leakage. We then evaluate state-of-the-art LLM-based decompilers and traditional tools on Decompile-Bench-Eval. The model trained on Decompile-Bench achieves a 20% improvement on re-executability rate over prior benchmarks. It also shows consistent gains across additional studied metrics, underscoring the effectiveness of our dataset for training LLM-based decompilers.

## Acknowledgments

This work is partially supported by the National Natural Science Foundation of China (No. 62372220), the Research Grants Council of the Hong Kong Special Administrative Region, China (Project No. PolyU/25200821), the Innovation and Technology Fund (Project No. PRP/047/22FX), and PolyU Internal Fund from RC-DSAI (Project No. 1-CE1E). This work is also partially supported by CCF-Kuaishou Large Model Explorer Fund (No. CCF-Kuaishou 2024013).

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

# A    Synthetic Datasets

## A.1    AnghaBench

AnghaBench [22], as noted above, injects synthetic types and dependencies so that each function can compile to an object file (not executable). As a result, both its source code and generated binaries differ remarkably from real-world programs, which limits its effectiveness when training decompilers on real-world binaries.

## A.2    Exebench

ExeBench [6] is the first publicly available dataset that pairs genuine C programs from GitHub with runnable I/O examples. It has inspired the development of many LLM-based decompilers [7, 89]. However, despite its GitHub origins, ExeBench's own analysis shows its executable codes exhibit substantially lower complexity than typical C code. We confirmed this by re-running their experiments using their original metrics as follows.

**Cyclomatic Complexity.** This metric measures the number of linearly independent paths through a program, offering a quantitative measurement of its control-flow complexity [4].

**Halstead's Metrics.** These metrics refer to a suite of measures introduced by Maurice Halstead that estimates development and maintenance effort based on the counts of distinct operators and operands in the code [35]. We use the Difficulty metric, it is the ratio of the number of unique operators to the total number of operators in the program.

We compute these metrics on two data splits: the executable subset of ExeBench (Executable Data in Figure 5), and a randomly selected collection from the full ExeBench to represent general GitHub functions (GitHub Data in Figure 5), following the original ExeBench protocol [6]. Our replicated results are summarized in Figure 5.

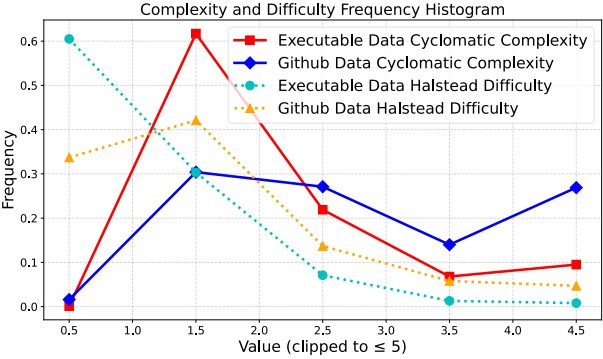

Figure 5: Code complexity for the executable subset of ExeBench and GitHub data. Note that the Halstead Difficulty is normalized by factor of 10 for better visulization.

Figure 5 shows that the ExeBench's executable subset with an average Cyclomatic Complexity of 2.1 and Halstead Difficulty of 10.5 is substantially simpler than the general GitHub data with an average of 3.6 and 16.3 in terms of these two metrics respectively. The distribution gap between the executable dataset and real-world code raises doubts about its value for training decompilers aiming at real-world applications.

# B    Data Filtering

As noted in Section 3.2, our Decompile-Bench, aka. the "clean data", retains only 2% of the original 100M "raw data". In Figure 6, we compare Cyclomatic Complexity and Halstead Difficulty between the raw and clean data. It shows that Decompile-Bench with an average Cyclomatic Complexity of 4.5 and Halstead Difficulty of 19.7 is significantly more complex than the unfiltered version with an average of 3.3 and 11.9 in terms of the two metrics respectively. Notably, comparing Figure 6 and

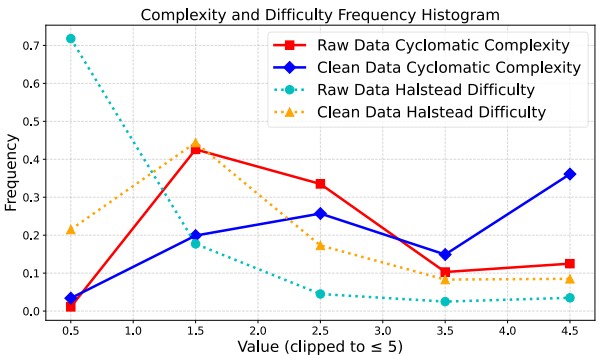

Figure 6: Code complexity for the raw data and clean data. The Halstead Difficulty is normalized by factor of 10 for better visulization.

Figure 5 reveals that the complexity distribution of our Decompile-Bench closely aligns with that of real-world GitHub code.

## C  Main Results for Other Metrics

We provide extra evaluation metrics for testing decompilers.

**Embedding Similarity.** Embedding similarity converts text into numerical vectors in a multi-dimensional space, where semantically similar texts have closer vectors. We embed the decompiled outputs and the source functions using a state-of-the-art model (i.e., CodeSage [93]) and compute cosine similarity between their vectors. Table 7 presents the results.

**Graph Similarity.** It is often argued that code is not just natural language [36], which has led to the development of various graph-based metrics to evaluate generated code. Specifically, CodeBLEU [70], though somewhat controversial [24], is one of the most commonly-used code metrics and also employed in decompilation [24]. CodeBLEU combines four evaluation aspects, i.e., BLEU score, weighted n-gram match, syntactic match via abstract syntax tree (AST) subtree overlap, and semantic match via data-flow graph comparison. Table 8 presents the results.

Table 7: Comparison of decompilers in terms of Codesage-embedding similarity.

| Codesage-Embd Similarity | HumanEval | | | | | MBPP | | | | | GitHub2025 | | | | |
|---|---|---|---|---|---|---|---|---|---|---|---|---|---|---|---|
| | O0 | O1 | O2 | O3 | AVG | O0 | O1 | O2 | O3 | AVG | O0 | O1 | O2 | O3 | AVG |
| GPT-4-1-mini | 56.04 | 40.13 | 43.26 | 35.96 | 43.85 | 48.67 | 39.11 | 43.49 | 33.16 | 41.11 | 40.30 | 32.45 | 31.61 | 32.11 | 34.12 |
| Ghidra | 29.72 | 28.07 | 30.22 | 26.11 | 28.53 | 29.32 | 28.65 | 27.83 | 26.36 | 28.04 | 55.75 | 49.02 | 54.45 | 56.55 | 53.94 |
| IDA | 27.88 | 23.76 | 22.10 | 20.76 | 23.63 | 27.31 | 22.92 | 21.43 | 19.96 | 22.91 | 54.49 | 51.25 | 51.08 | 51.90 | 52.36 |
| LLM4Decompile-End | 48.09 | 42.50 | 41.86 | 41.99 | 43.61 | 44.54 | 40.90 | 40.30 | 39.44 | 41.30 | 57.60 | 57.35 | 55.82 | 55.49 | 56.57 |
| LLM4Decompile-DCBench | 59.01 | 51.10 | 50.86 | 50.17 | 52.79 | 53.61 | 47.71 | 46.73 | 45.73 | 48.45 | 62.30 | 64.84 | 64.08 | 60.73 | 62.98 |

Table 8: Comparison of decompilers in terms of CodeBLEU rates.

| CodeBLEU | HumanEval | | | | | MBPP | | | | | GitHub2025 | | | | |
|---|---|---|---|---|---|---|---|---|---|---|---|---|---|---|---|
| | O0 | O1 | O2 | O3 | AVG | O0 | O1 | O2 | O3 | AVG | O0 | O1 | O2 | O3 | AVG |
| GPT-4-1-mini | 36.98 | 28.11 | 29.13 | 24.27 | 29.62 | 40.43 | 31.99 | 34.68 | 27.78 | 33.72 | 18.33 | 17.55 | 17.24 | 16.85 | 17.49 |
| Ghidra | 23.27 | 22.34 | 22.05 | 22.70 | 22.59 | 26.02 | 24.42 | 24.28 | 24.67 | 24.86 | 20.98 | 22.56 | 22.48 | 21.61 | 21.91 |
| IDA | 24.78 | 22.43 | 21.84 | 22.16 | 22.80 | 27.45 | 24.05 | 23.88 | 24.71 | 25.02 | 20.99 | 22.38 | 21.91 | 21.26 | 21.64 |
| LLM4Decompile-End | 32.46 | 27.71 | 26.67 | 26.10 | 28.24 | 34.66 | 29.33 | 28.73 | 28.36 | 30.27 | 21.80 | 22.15 | 21.69 | 21.06 | 21.68 |
| LLM4Decompile-DCBench | 39.09 | 29.97 | 30.09 | 29.96 | 32.28 | 42.35 | 35.80 | 35.02 | 28.36 | 35.35 | 25.41 | 25.39 | 24.57 | 22.86 | 24.56 |

As shown in Tables 7 and 8, we observe the same trends as in re-executability, R2I, and edit similarity. LLM4Decompile-DCBench achieves an averaged 16.6% and 14.8% improvement on embedding similarity and CodeBLEU over LLM4Decompile on all three test sets. In summary, the LLM4Decompile-DCBench, trained on Decompile-Bench, delivers the strongest performance, underscoring the value of using real-world data to train an LLM-based decompiler.

We utilized GPT-4 as an expert judge to score the outputs from each decompiler on GitHub2025 with a scale of 1 (poor) to 100 (excellent) across three qualitative areas, i.e., variable name recovery, control flow clarity, and type reconstruction with the prompt as shown in Figure 7.

```
You are an expert in reverse-engineering and decompiler evaluation.  I will give you a decompiled code snippet; your
job is to evaluate it on three criteria:

    1. variable_naming: How well the decompiler recovered meaningful variable names.
    2. control_flow: How faithfully complex control-flow constructs (loops, branches, gotos) have been reconstructed.
    3. type_recovery: How accurately types (primitives, structs, pointers, arrays, etc.) were inferred.

    For each criterion:
    • Assign an integer score from 1 (very poor) to 100 (excellent).
    • Provide a one- or two-sentence rationale.

    Produce only a single JSON object, with exactly these fields:

    {
    "variable_naming": {
        "score": <int>,
        "rationale": "<string>"
    },
    "control_flow": {
        "score": <int>,
        "rationale": "<string>"
    },
    "type_recovery": {
        "score": <int>,
        "rationale": "<string>"
    }
    }

    Do not include any extraneous keys and directly output the result without any explanation.
    The source code: {source code}.
    Now evaluate this snippet: {decompiled code}
```

Figure 7: GPT-Judge prompt for evaluation on variable name recovery, control flow clarity, and type reconstruction.

Table 9: GPT-Judge on variable naming, control flow and type recovery using Github2025.

| GPT-Judge | Varaiable Naming | | | | | Control Flow | | | | | Type Recovery | | | | |
|---|---|---|---|---|---|---|---|---|---|---|---|---|---|---|---|
| | O0 | O1 | O2 | O3 | AVG | O0 | O1 | O2 | O3 | AVG | O0 | O1 | O2 | O3 | AVG |
| GPT-4-1-mini | 48.99 | 42.24 | 43.07 | 39.98 | 43.57 | 63.25 | 50.09 | 50.41 | 50.10 | 53.46 | 55.69 | 45.18 | 46.75 | 44.93 | 48.14 |
| IDA | 33.66 | 27.16 | 29.49 | 28.99 | 29.83 | 63.28 | 59.42 | 60.35 | 60.62 | 60.92 | 63.53 | 60.37 | 62.29 | 61.67 | 61.97 |
| LLM4Decompile-End | 64.15 | 63.48 | 62.39 | 63.84 | 63.47 | 73.75 | 73.49 | 73.61 | 74.65 | 73.88 | 76.47 | 77.82 | 78.98 | 77.42 | 77.67 |
| LLM4Decompile-DCBench | 76.38 | 77.18 | 77.53 | 76.69 | 76.95 | 83.61 | 85.13 | 85.56 | 84.87 | 84.79 | 80.13 | 82.26 | 82.22 | 81.55 | 81.54 |

Table 9 summarizes the GPT evaluation results. IDA struggles with semantic richness, achieving a variable naming score that is only 38.8% of our model's. GPT fails on logical integrity, on control flow recovery, it scores only 63.1% of our model's. LLM4Decompile-DCBench significantly outperform all baselines. Notably, it improves upon the strong LLM4Decompile-End base model by 21.2% in name recovery, 14.7% in control flow, and 5.0% in type reconstruction. In summary, this comprehensive qualitative analysis demonstrates that while traditional tools preserve logic at the cost of readability, and general LLMs sacrifice logic for readability, our approach successfully achieves both. This makes our model's output significantly more valuable for the practical tasks that reverse engineers and vulnerability researchers care about.

**Other Metrics**  Beyond the metrics we cover, prior work has also used techniques such as symbolic execution [13], CodeAlign [24], D-Helix [97], human judgments [61], and GPT evaluation [76, 60, 86]. We leave these evaluation in future work due to their availability, heavy manual effort, or costs. Likewise, name-recovery or type-focused scores, such as exact match [51, 16] or variable similarity [17], are left out. In particular, these metrics only assess isolated elements of the decompiled output, whereas our goal is to evaluate correctness and readability for the whole function.

## D   Binary-Source Search

Our binary–source benchmark is directly applicable to binary-to-source search, i.e., a core task in third-party library detection, software composition analysis, etc [43, 90]. We train a binary–source embedding model on 10% of our data using a conventional contrastive loss [93, 98]. In particular, we strip out all assembly function names to prevent trivial matches.

Table 10: Binary-source search results on Github data in Decompile-Bench-Eval.

| Optimization Level | O0 | O1 | O2 | O3 | AVG |
|---|---|---|---|---|---|
| Recall@1 | 32.5 | 24.7 | 28.1 | 22.8 | 27.0 |

For evaluation, we use the GitHub2025 dataset and split it into (1) a reference corpus of 17 K source functions as our "search database", and (2) a set of 66 K binary (assembly) functions, each of which acts as a query. Each query is embedded and used to retrieve its closest matches by cosine similarity from the source-function database. We then measure retrieval quality in terms of recall@k, i.e., the fraction of queries whose true match appears within the top k retrieved results. Table 10 reports recall@1 results using just 10% of the training data. We attain 27.0% recall@1, on par with the results reported in state-of-the-art binary-source software component analysis tool BinaryAI [43], which achieves 22.4% recall@1 under a larger search space of 32K. The promising results show that Decompile-Bench can significantly assist researchers with binary–source search tasks and offers potential applications in third-party library detection and software component analysis [43, 90, 91].

# E   Data Quality Analysis

To verify the correctness of binary-source matching, we conducted a two-part evaluation, i.e., a direct, qualitative analysis and an indirect, empirical validation through downstream task performance.

## E.1   Direct, Qualitative Evaluation

A key challenge is that a ground-truth one-to-one mapping often does not exist. Therefore, we performed a rigorous qualitative evaluation, which we consider is rather fit and meaningful for this task. We manually inspected a random sample of 1,000 pairs from our dataset. Our analysis confirmed that Algorithm1 works as intended, with the observed "noise" falling into two distinct and expected categories as follows.

**Negligible Parser Noise (<0.1% of cases)**. In extremely rare instances, the source code parser (tree-sitter) makes minor segmentation errors (e.g., with nested functions). These cases are statistically insignificant and unlikely to impact training effectiveness.

**Compiler-Induced Scope Mismatches (~25% of cases)**. A far more common scenario arises from aggressive compiler optimizations, particularly function inlining. This results in a binary function that correctly contains code from its primary source function plus code from other inlined functions. This is not an algorithmic error but an authentic and unavoidable artifact of real-world compilation. We consider these pairs to be valuable and necessary training data that guides the model to handle the complexities of optimized binaries.

## E.2   Indirect, Empirical Validation

Data quality could be directly reflected by its effectiveness in training. LLM4Decompile-DCBench, which was fine-tuned on the dataset generated by Algorithm 1, achieved a 20% relative performance gain over its base model. Such a significant improvement would be highly unlikely if the binary-source pairs were noisy or incorrectly matched. This strong downstream result serves as powerful empirical evidence that the pairs generated by our algorithm are of high quality and correctness.

# F   Adaptability of CTF Framework

We designed our pipeline to be modular, which makes our framework highly adaptable. First, for other compiled languages, the core requirement is to adapt the compilation stage (inherently language dependent) and the DWARF parser for the target language's specific conventions. For instance, The Rust language would require minimal changes, as its DWARF debug format is highly similar to C/C++ [78]. The GO language would require a custom DWARF parser due to its unique format [33], but our core tracing and alignment algorithms would remain applicable.

Second, regarding build systems for C/C++, we successfully added support for projects using Makefiles, which, along with CMake, represent the vast majority of C/C++ build systems. We released 643K new data pairs (condensed from 12M pairs) from 2,600 permissive C/C++ projects.

## G   Discussion on Unit Test on Real-World Projects

Re-executability is a key challenge in decompilation evaluation for real-world projects, and we share the view that it remains a significant challenge for real-world projects.

Re-executing GitHub test cases is notably absent from other recent, prominent decompilation works (e.g., LLM4Decompile [76], Nova [44], Idioms [25]) and evaluation benchmarks (e.g., Assemblage [57], BinBench [19]). Resolving the difficulty of such an evaluation may deserve another research paper.

In our own work, we invested significant effort in attempting to build a re-executability benchmark. Our investigation revealed two fundamental and currently prohibitive obstacles. The first is pervasive data leakage in existing test suites. Specifically, a common approach would be to leverage existing projects with built-in test suites, such as Defects4C [1] benchmark, or well-tested codebases like the Git or FFmpeg (which supports fuzzing to generate inputs). However, these famous and widely-studied codebases are most likely part of the pre-training corpora for LLMs like GPT/Claude, and are included in fine-tuning datasets for specialized models like LLM4Decompile/Idioms. Evaluating such data would produce misleadingly high scores due to data leakage, failing to measure a model's true generalization capabilities. The second obstacle is the prohibitive cost of manual, leak-free benchmark construction. Specifically, a more rigorous but extremely labor-intensive alternative is to assemble a benchmark of truly unseen, real-world projects. We attempted this manually and shared our experiences as follows.

**Project discovery (10 min/repo).** We search GitHub for post-2025 repositories whose top-level folders or README mention `test`, `demo` or `sample` and manually verify the presence of sample code and attempt compilation (which often fails).

**Sample execution (30–60 min/repo).** We understand the samples, run them, and capture the inputs and outputs (execution frequently errors out).

**Trace binary (30–60 min/repo).** We map each sample's function calls back to their source definitions and match those to the compiled binary.

**End-to-end automation (60+ min/repo).** We build a Docker environment to reproducibly compile the project with decompiled code, run the samples, and report pass/fail states (i.e., engineering challenge).

A single author spent over three hours attempting to complete just the first three steps for one project, encountering multiple failures before finding a suitable candidate. However, we still could not stably automate the compilation and execution step (i.e., end-to-end automation), as it is an even greater engineering challenge. It is worth noting that while unit test generation techniques can assist in creating test cases during the project discovery phase, the subsequent challenges of automatic compilation, test execution, and result collection remain significant obstacles.

While we believe building a real-world, leakage-free, and executable evaluation benchmark is a critical direction for the decompilation community, it goes beyond the scope of this paper.

## H   Limitations

Due to resource constraints, we trained only 10% of the Decompile-Bench on a 1.3B model, which took around 4 hours on an A800 × 8 setup. While a larger model trained on the full-scale data would likely provide a comprehensive understanding of Decompile-Bench, it would require several months of training, i.e., a computation cost that exceeds our budget. Moreover, decompilation at the project scope would ideally be more effective. While Decompile-Bench records the project-level meta data, this approach is heavily limited by the model's sequence length and available computational power, rendering it both extremely expensive and nearly unfeasible. Finally, legal concerns remain a critical issue, as developing an effective decompiler might require to include non-permissive or commercial

data, potentially violating legal frameworks and leading to misuse. Therefore, Decompile-Bench strictly follow permissive license and discard large amount of non-permissive GitHub projects.

