# OpenReview forum: "Decompile-Bench: Million-Scale Binary-Source Function Pairs for Real-World Binary Decompilation"
_NeurIPS.cc/2025/Datasets_and_Benchmarks_Track — NeurIPS 2025 Datasets and Benchmarks Track poster_

### Official Review · Reviewer_jJhk · 2025-06-07

**Rating:** 4
**Confidence:** 2

**Summary:**

This paper presents DecompileBench, a benchmark for evaluating the ability of LLMs to decompile binaries/assembly codes to source codes. The authors present two datasets, a training set DecompileBench (with 2M source-code-binary pairs) and a test dataset (containing manually curated HumanEval/MBPP source code-binary pairs and a Github 2025 source code-binary pair). The authors prepare the datasets with some good filtering techniques to mitigate duplication, trivial codes, and lack of diversity, and present multiple metrics for evaluation. Experimental results show that fine-tuning with the training set DecomplieBench leads to better metrics on the test set DecompileBench-Eval. In addition, via ablation studies, the authors show that the proposed filtering techniques lead to a performance improvement on the test set.

**Additional Feedback:**

NA

**Dataset Code Accessibility:**

Partly

**Dataset Code Comments:**

The datasets and evaluation pipeline is given in the associated url. It would be good if the authors can provide the training pipeline as well.

**Ethical Considerations:**

No, there are no or only very minor ethics concerns

**Final Justification:**

The rebuttal is informative and provide me with additional insights on this paper. I recommend accept.

**Limitations Weaknesses:**

1. The evaluation process may be biased towards the proposed methods. It is possible that, since the test datasets (especially the Github2025) undergo similar processing as the training dataset, it is possible that the models trained with DecompileBench is not stronger, but just fits the data distribution better. To answer the question, the authors may test the trained models on other datasets in Figure 1.
2. The algorithm 1 of pairing binary with source functions is not evaluated. The authors claim that it 'yields precise binary–source function pairs', but the claim is not backed up with experiments.
3. I am not an expert in decompilation, so I have some questions regarding the metrics.
- Regarding the re-executability, I am not sure where the test cases come from. Do they come from the github repos themselves or the authors generate them?
- Regarding the edit distance, I am not sure whether some changes in naming would affect this (e.g. name of variables, functions, etc. ).
- The authors may need to specify the platform of the binary/assembly (e.g. x86 or arm) and the source code (e.g. C++11, C++19, etc).
4. Since Claude is strong in software engineering, it is better if the authors can test Claude models as well.
5. Since you only train your models on 10% of the entire DecompileBench, the training set is ~200K, which is similar to Idioms in Related Work. Although the pre-processing of idioms is not clear, it is good if the authors do a best-effort approach to try to run it.

**Strengths Contributions:**

1. The training dataset DecompileBench comes from real-world Github code and comes with large scale (2M source code-binary pairs).
2. The training dataset undergoes significant pre-processing and filtering, many of which will prove to be useful to future research in this field. For example, the automatic compilation makes it easier for users to re-use and re-execute the code in a controlled manner. The binary-source linking makes the code and binaries cleaner and more easily navigated. The filters de-duplicates the data and cleans up trivial functions. These are heuristics very specific to the field of coding and software engineering, and I believe they would be of interest to a wider range of researchers. Also, these techniques turn out to be helpful when the dataset is used for training.
3. Evaluations are done in an extensive manner. The authors perform experiments under various optimization levels (O1 to O3) and use multiple metrics for evaluations. Ablation studies are also done to show the effectiveness of data filtering.

---

> ### Author Rebuttal · Authors · 2025-07-31
>
> Thank you for the encouraging feedback and helpful suggestions. Please find the itemized responses below.
>
> **Q1,Q4,Q5:**  ...test the models on other datasets...test Claude models...try to run idioms.
>
> **A1:** Thank you for these critical suggestions. We have added two key baselines (Claude-Sonnet-4-reasoning and Idioms-1.3B) and evaluated all models on an independent benchmark, ProRec [1], a NeurIPS 2024 dataset, the only available dataset with assembly-source pairs. We also include a model scaling experiment to demonstrate the data efficiency of DecompileBench.
>
> 1.Generalization on an Independent Benchmark (ProRec):
> | Model/Metric               | Edit Sim | R2I   |
> |----------------------------|----------|-------|
> | GPT-4.1-mini               | 34.74    | 55.01 |
> | IDA                        | 27.24    | 38.35 |
> | LLM4Decompile-End-1.3B     | 34.26    | 57.49 |
> | Idioms-1.3B                | 36.03    | 64.86 |
> | LLM4Decompile-DCBench-1.3B | 38.85    | 65.73 |
> | LLM4Decompile-DCBench-6.7B | 45.23    | 66.15 |
> | Claude-Sonnet-4-reasoning  | 41.99    | 57.38 |
>
> On this new dataset, the performance trend remains consistent with our other evaluations. The base model, LLM4Decompile-End, is outperformed by Idioms, but our model, LLM4Decompile-DCBench, performs the best among the 1.3B models. This consistency on an external dataset is a strong evidence against overfitting.
>
> Interestingly, while Claude4 excels at Edit Sim, it performs poorly on R2I. A manual inspection revealed that Claude's outputs are often verbose, with an average line count 15% longer than our model's. This verbosity penalizes it on length-sensitive metric, highlighting a key difference in metric's preference.
>
> 2.Comparison with SOTA Baselines (Claude4 & Idioms):
> | HumanEval                  | O0    | O1    | O2    | O3    | Average |
> |----------------------------|-------|-------|-------|-------|---------|
> | GPT-4.1-mini               | 21.95 | 11.58 | 10.07 | 10.06 | 13.42   |
> | IDA                        | 18.60 | 19.81 | 17.69 | 16.77 | 18.22   |
> | LLM4Decompile-End-1.3b     | 26.22 | 12.81 | 14.03 | 13.42 | 16.22   |
> | Idioms-1.3B                | 30.56 | 16.10 | 12.63 | 12.36 | 17.91   |
> | llm4decompile-DCBench-1.3B | 33.23 | 18.60 | 16.47 | 15.24 | 20.89   |
> | llm4decompile-DCBench-6.7B | 61.59 | 30.18 | 34.15 | 32.01 | 39.48   |
> | Claude4-sonnet-reasoning   | 65.85 | 42.68 | 39.63 | 39.02 | 46.79   |
>
> Claude4 achieves the highest re-executability rate (46.8%), demonstrating the power of large-scale commercial LLMs.
> Idioms shows promise on O0 optimizations (30% re-executability) but its performance degrades sharply on higher optimizations like O2/O3 (~12%). This suggests its training data may lack sufficient examples of aggressively optimized code, validating the effectiveness of our Automatic Compilation technique (Section 3.1) which ensures such cases are well-represented.
>
> 3.Data Efficiency and Scaling:
> We trained a larger 6.7B parameter model on 10% of our data.
> This model achieves a 39.5% re-executability rate, which is highly competitive with Claude4's 46.8%. It indicates that a model trained on a small fraction of our high-quality data can approach the performance of a massive commercial LLM trained on trillions of tokens.
>
> In summary, these new experiments demonstrate that our model generalizes to external datasets, our data generation process creates a more robust training set, and our DCBench is highly efficient for training powerful decompilers.
>
> (Note: Due to space constraints, **full results over three metrics on four datasets are available on the anonymous GitHub page.**)
>
> [1] Su, Zian, et. al,. "Source code foundation models are transferable binary analysis knowledge bases." NeurIPS2024.
>
>
> **Q2:**  ...algorithm 1 is not evaluated...
>
> **A2:** Thank you for raising this important concern. To affirm the correctness of Algorithm 1, we conducted a two-part evaluation:
>
> 1.Direct Qualitative Evaluation:
>
> We manually inspected a random sample of 1,000 pairs from our dataset, which we argue is the most meaningful method for this task as the "ground truth" for the binary-source mapping does not exist. Our analysis confirmed that Algorithm 1 works as intended, with the observed "noise" falling into two categories:
>
> **Negligible Parser Noise (<0.1% of cases):** In extremely rare instances, the source code parser (tree-sitter) makes minor segmentation errors (e.g., with nested functions). These cases are statistically insignificant and unlikely to impact training effectiveness.
>
> **Compiler-Induced Scope Mismatches (~25% of cases):** A far more common scenario arises from aggressive compiler optimizations, particularly function inlining. This results in a binary function that **correctly** contains code from its primary source function plus code from other inlined functions. This is not an algorithmic error but an authentic and unavoidable artifact of real-world compilation. We consider these pairs to be **valuable and necessary training data** that teaches the model to handle the complexities of optimized binaries.
>
> 2.Indirect Empirical Validation:
>
> The ultimate test of the data's quality is its **effectiveness in training**. Our model, LLM4Decompile-DCBench, which was fine-tuned on the dataset generated by Algorithm 1, achieved a 20% relative performance gain over its base model. This strong downstream result serves as powerful empirical evidence that the pairs generated by our algorithm are of high quality and correctness.
>
>
> **Q3-1:** ...re-executability, where the GitHub test cases come from...
>
> **A3-1:** Thanks for the question. Creating the test cases is a well-recognized challenge in the field, which is why **re-executability GitHub test cases are notably absent from other recent decompilation works (e.g., LLM4Decompile, Nova, Idioms) and evaluation benchmarks (e.g., Assemblage, BinBench)**. The difficulty of creating such an evaluation may worth another research paper.
>
> In our own work, we invested significant effort in attempting to build a re-executability benchmark. Our investigation revealed two fundamental and currently prohibitive obstacles:
>
> **Pervasive data leakage in existing test suites:**
> First, a common approach would be to leverage existing projects with built-in test suites, such as Defects4C benchmark, or well-tested codebases like the Git or FFmpeg (supports fuzzing to generate inputs). However, these famous and widely-studied codebases are almost certainly part of the pre-training corpora for LLMs like GPT/Claude, and are already included in fine-tuning datasets for specialized models like LLM4Decompile/Idioms. Evaluating on such data would violate the principles of fair evaluation and fail to measure a model's true generalization capabilities due to data leakage.
>
> **Prohibitive Cost of Manual, Leak-Free Benchmark Construction:**
> Second, a more rigorous but extremely labor-intensive alternative is to assemble a benchmark of truly unseen, real-world projects. We attempted this manually and shared our experiences:
>
> 1: Project discovery (∼10 min/repo). Search GitHub for post-2025 repositories whose top-level folders or README mention “test”, “demo” or “sample”. Manually verify the presence of sample code and attempt compilation (often fails).
>
> 2: Sample execution (30–60 min/repo). Understand the samples and run them, capture the inputs and outputs (execution frequently errors out).
>
> 3: Trace binary (30–60 min/repo). Map each sample’s function calls back to their source definitions, and match those to the compiled binary.
>
> 4: End-to-end automation (60+ min/repo). Build a Docker environment to reproducibly compile the project with decompiled code, run the samples, and report pass/fail status (engineering challenge)
>
> A single author spent over three hours attempting to complete just the first three steps, encountering multiple failures before finding a suitable candidate. However, we still could not stably automate the compilation and execution step (Step 4), as it is an even greater engineering challenge.
> It is worth noting that while unit test generation techniques can assist in creating test cases during the project discovery phase (Step 1), the subsequent challenges of automatic compilation, test execution, and result collection remain significant obstacles.
>
> While we believe building a real-world, leakage-free, executable evaluation benchmark is a critical direction, it was beyond the scope of this work and we leave it as important future work.
>
> **Q3-2:** ...edit distance, whether changes in naming would affect this...
>
> **A3-2:** Thanks for the question. Yes, the edit distance score is directly affected by changes to variable and function names. Edit distance measures the minimum number of single-character edits (insertions, deletions, or substitutions) required to transform one string into another. Because variable and function names are part of the code's text, any change to them is penalized at a character level.
>
> We recognize that purely text-based metrics like edit distance can penalize semantically equivalent but syntactically different code and thus have limitations in decompilation evaluation. However, because it remains a **standard, widely-adopted measure in prior work like Slade and LLM4Decompile**, we include it in our evaluation for the sake of comparability.
>
> **Q3-3:** ...specify the platform of the binary/assembly...
>
> **A3-3:** Thank you for catching that. We compiled our binaries on Ubuntu 20.04 targeting the Linux x86_64 architecture, using Clang 19 with the C++17 standard.
>
> **Q6:** ...provide the training pipeline...
>
> **A6:** Thank you for the suggestion. We have published two full training scripts for some time:
>
> Colossal-AI–based training script (released Aug 2024).
>
> LLaMA-Factory–based training script (released May 2025).
>
> We have now merged both scripts to the "train" directory of the anonymous GitHub repo.

---

> > ### Comment · Reviewer_jJhk · 2025-08-06
> >
> > Thank you for your rebuttal. The rebuttal answers my questions and concerns very well.
> >
> > After reading the rebuttal, I am confident that the paper would be useful and of interest to the community. I recommend accept.

---

> > > ### Author Response · Authors · 2025-08-07
> > >
> > > Thank you for your positive feedback. We are glad to hear that our rebuttal successfully addressed your questions and concerns. We appreciate your time and constructive comments, which have helped us improve the paper.

---

### Official Review · Reviewer_X9M1 · 2025-06-28

**Ethics Flags:** Data privacy, copyright, and consent,…
**Rating:** 5
**Confidence:** 1

**Summary:**

This paper introduces Decompile-Bench, a large-scale, open-source benchmark for evaluating LLM-based decompilers. The benchmark comprises two million high-quality binary–source function pairs distilled from over 100 million raw pairs addressing the lack of a large open-source datasets for LLM decompilers.

The authors also propose a pipeline (CTF framework) to automate project builds, that they used to build their benchmark and provide some statics on Decompile-Bench. Moreover, they extend their benchmark proposing Decompile-Bench-Eval (for testing purposes) by including binaries crafted using popular code completion benchmarks and binaries compiled from Github repositories.

Finally, models trained on Decompile-Bench are evaluated using standard metrics on Decompile-Bench-Eval and are also compare with models trained on other datasets. The empirical results show that decompilers fine-tuned on Decompile-Bench achieve a 20% improvement compared to models trained on other datasets.

**Dataset Code Accessibility:**

Yes

**Dataset Code Comments:**

The code provided in the GitHub link is well documented and easy to use, providing clear instructions for reproducing most of the experiments. However, I could not find the training script for training the LLM models (but trained models are provided).

I did not find a detailed description of the datasets on the Hugging Face page, although the datasets are well described in both the paper and the accompanying GitHub repository. Including a clear and concise dataset description (indicating for instance the meaning of columns as done in the Github page) directly on the Hugging Face page would be beneficial for users who discover the dataset independently, without referring to the paper or GitHub documentation. This would also improve accessibility and encourage broader adoption.

Both training and test datasets can be conveniently downloaded from Hugging Face using the following scripts:

```
from datasets import load_dataset

ds = load_dataset("LLM4Binary/decompile-bench")
```

and

```
from datasets import load_dataset

ds = load_dataset("LLM4Binary/decompile-eval")
```

**Ethical Comments:**

There are a few potential ethical concerns associated with releasing large-scale decompilation datasets, including the risk of violating software licenses, unintentionally enabling reverse engineering of proprietary or sensitive software, and the possibility of dual-use (e.g., misuse for malicious purposes such as malware analysis).

However, the authors have taken steps to responsibly address these issues. They clarify that the dataset is built exclusively from permissively licensed GitHub repositories, and they make efforts to mitigate data leakage by including evaluation data compiled from repositories published after 2025. These measures help ensure responsible dataset construction and usage

Additionally, they explicitly state that commercial software remains protected due to prevalent use of obfuscation techniques, which prevent effective decompilation, as confirmed in prior research. Finally, the benchmark is released for academic and educational use, with the aim of advancing research in binary analysis and reverse engineering education.

**Ethical Considerations:**

Yes, there are ethics concerns that require attention by the authors

**Final Justification:**

After reading the comments from other reviewers and the authors' rebuttal, I believe the authors have addressed the main concerns raised by both myself and others. As I am not an expert in this specific field, I have decided to maintain my original score of 5 (Accept).

**Limitations Weaknesses:**

Unfortunately, I am not an expert in the domain, so I did not identify major limitations beyond those already acknowledged by the authors in the "Limitations" section (reason why my confidence score is 1). One minor issue I observed is the absence of training scripts in the provided GitHub repository. Including these scripts would enhance the reproducibility of the results and make it easier for other researchers to fine-tune or extend the models using the proposed benchmark. Providing complete training pipelines alongside the dataset and evaluation tools would strengthen the usability and long-term impact of the work.

**Strengths Contributions:**

The paper is well-written and easy to follow, with a clear structure that effectively communicates the motivation, methodology, and contributions. To the best of my knowledge, this is the first open-source, large-scale benchmark specifically targeting real-world binary–source function pairs for decompilation.  The authors propose a carefully designed data collection and filtering pipeline that ensures the quality and accuracy of the dataset despite challenges like compiler optimizations and function inlining. Overall, the paper offers a valuable resource for the community by providing this benchmark accompanied by strong empirical results (e.g., a 20% improvement in re-executability) that demonstrate its effectiveness.

---

> ### Author Rebuttal · Authors · 2025-07-31
>
> Thank you for the encouraging feedback and helpful suggestions. Please find the itemized responses to the questions below.
>
> **Q1:** One minor issue I observed is the absence of training scripts in the provided GitHub repository.
>
> **A1:** Thank you for catching that.
>
> To clarify, we have published two full training scripts for some time:
>
> Colossal-AI–based training script (released Aug 2024). We leverage Colossal-AI to have full control over every step—custom dataloaders, loss functions, etc.
>
> LLaMA-Factory–based training script (released May 2025). We use LLaMA-Factory for a streamlined workflow, so others can train models without worrying about data loading or loss-calculation details.
>
> We have now merged both scripts to the "train" directory of the anonymous repo.
>
> **Q2:** There are a few potential ethical concerns associated with releasing large-scale decompilation datasets...However, the authors have taken steps to responsibly address these issues...Additionally, they explicitly state that commercial software remains protected due to prevalent use of obfuscation techniques...Finally, the benchmark is released for academic and educational use.
>
> **A2:** We are grateful for your recognition of the specific measures we implemented to ensure the responsible construction and release of our decompilation benchmark. As you noted, we were meticulous in building the dataset exclusively from permissively licensed open-source repositories, mitigating potential data leakage with a time-sensitive evaluation set, and clarifying the practical limitations of decompilation on obfuscated commercial software.
>
> Your understanding of our commitment to fostering academic and educational advancement in binary analysis and reverse engineering is greatly appreciated. We believe that by taking these deliberate steps, we can share a valuable resource with the research community while upholding our ethical responsibilities.
>
> **Q3:** I did not find a detailed description of the datasets on the Hugging Face page, although the datasets are well described in both the paper and the accompanying GitHub repository.
>
> **A3:** Thank you for the suggestion. We’ve updated the details on our Hugging Face page.

---

> > ### Comment · Reviewer_X9M1 · 2025-08-02
> >
> > I thank the authors for their response. They have addressed my concerns. After reading the other reviews and author's  answers, I decided to keep my score (5: Accept). Since I am not an expert in this field, I have decided to keep my confidence score of 1.

---

> > > ### Author Response · Authors · 2025-08-02
> > >
> > > Thank you very much for your time and for reviewing our work. We appreciate your positive feedback and are glad to hear that we were able to address your concerns.

---

### Official Review · Reviewer_y9L1 · 2025-07-03

**Rating:** 5
**Confidence:** 4

**Summary:**

This paper addresses a critical gap in decompilation research: the lack of a large-scale, realistic benchmark for training and evaluating LLM-based decompilers. The authors argue that existing datasets are often synthetic, small-scale, or provide only fragmented mappings, which limits their real-world applicability.

This paper creates and releases two datasets:
* Decompile-Bench, a new benchmark of two million function-level binary-source pairs derived from permissively licensed GitHub projects. The authors introduce a robust.
* Decompile-Bench-Eval, a leakage-resistant test set using HumanEval, MBPP, and recent GitHub projects.

Fine-tuning an existing model on Decompile-Bench demonstrates a significant, ~20% relative improvement in re-executability compared to prior benchmarks.  This work presents a valuable and well-engineered resource that convincingly demonstrates the benefit of large-scale, high-quality, real-world data for advancing decompilation technology.

**Dataset Code Accessibility:**

Yes

**Dataset Code Comments:**

The dataset is released on HuggingFace, and the source code for evaluation is also released in a GitHub repository.

**Ethical Considerations:**

No, there are no or only very minor ethics concerns

**Final Justification:**

After reading the authors' response, I am overall satisfied with this paper and the benchmark and I will keep my rating.

**Limitations Weaknesses:**

1. The GitHub2025 evaluation dataset only supports evaluation metrics such as R2I readability and edit similarity score. It still remains challenging to evaluate the re-executability of the decompilation of real-world projects.

**Strengths Contributions:**

1. A large-scale dataset, Decompile-Bench, containing more than 2 million function-level binary-source pairs. This large dataset can be used to train or fine-tune models for developing binary code decompilation tools.

2. An evaluation benchmark and several evaluation metrics, including re-executability, R2I readability, and edit similarity.

---

> ### Author Rebuttal · Authors · 2025-07-31
>
> Thank you for the encouraging feedback and helpful suggestions. Please find the responses to the re-executability challenge below.
>
> **Q:** The GitHub2025 evaluation dataset only supports evaluation metrics such as R2I readability and edit similarity score. It still remains challenging to evaluate the re-executability of the decompilation of real-world projects.
>
> **A:** Thank you for raising this crucial point. We agree that re-executability is a key challenge in decompilation evaluation for real-world projects, and we share the view that it remains a significant challenge for real-world projects.
>
> This is a well-recognized challenge in the field, which is why **re-executability GitHub test cases are notably absent from other recent, prominent decompilation works (e.g., LLM4Decompile, Nova, Idioms) and evaluation benchmarks (e.g., Assemblage, BinBench).** The difficulty of creating such an evaluation may worth another research paper.
>
> In our own work, we invested significant effort in attempting to build a re-executability benchmark. Our investigation revealed two fundamental and currently prohibitive obstacles:
>
> **Pervasive data leakage in existing test suites:**
>
> First, a common approach would be to leverage existing projects with built-in test suites, such as Defects4C [1] benchmark, or well-tested codebases like the Git or FFmpeg (supports fuzzing to generate inputs). However, these famous and widely-studied codebases are almost certainly part of the pre-training corpora for LLMs like GPT/Claude, and are included in fine-tuning datasets for specialized models like LLM4Decompile/Idioms. Evaluating on such data would produce misleadingly high scores due to **data leakage**, violating the principles of fair evaluation and failing to measure a model's true generalization capabilities.
>
> **Prohibitive Cost of Manual, Leak-Free Benchmark Construction:**
>
> Second, a more rigorous but extremely labor-intensive alternative is to assemble a benchmark of truly unseen, real-world projects. We attempted this manually and shared our experiences:
>
> 1: Project discovery (∼10 min/repo). Search GitHub for post-2025 repositories whose top-level folders or README mention “test”, “demo” or “sample”. Manually verify the presence of sample code and attempt compilation (often fails).
>
> 2: Sample execution (30–60 min/repo). Understand the samples and run them, capture the inputs and outputs (execution frequently errors out).
>
> 3: Trace binary (30–60 min/repo). Map each sample’s function calls back to their source definitions, and match those to the compiled binary.
>
> 4: End-to-end automation (60+ min/repo). Build a Docker environment to reproducibly compile the project with decompiled code, run the samples, and report pass/fail status (engineering challenge).
>
> A single author spent over three hours attempting to complete just the first three steps for one project, encountering multiple failures before finding a suitable candidate. However, we still could not stably automate the compilation and execution step (Step 4), as it is an even greater engineering challenge.
> It is worth noting that while unit test generation techniques can assist in creating test cases during the project discovery phase (Step 1), the subsequent challenges of automatic compilation, test execution, and result collection remain significant obstacles.
>
> While we believe building a real-world, leakage-free, executable evaluation benchmark is a critical direction for the decompilation community, it was beyond the scope of this work and we leave the construction of such a benchmark as important future work.
>
> [1] Wang, Jian Jornbowrl, Xiaofei Xie, Shangqing Liu, Jiaolong Kong, Jiongchi Yu, and Yi Li. "Defects4C: Benchmarking C/C++ Faults to Assess LLM-Based Program Repair."

---

### Official Review · Reviewer_xHmv · 2025-07-03

**Rating:** 5
**Confidence:** 1

**Summary:**

This paper tackles a longstanding gap in the decompilation and binary analysis field by introducing Decompile-Bench, an impressively large and practical benchmark. The dataset contains two million function-level binary-to-source mappings extracted from real C/C++ projects on GitHub, representing a huge step up from the synthetic and contest-style benchmarks we’ve been stuck with. The authors also propose a pipeline (CTF) for reliably building, tracing, and filtering the data. Their evaluation shows that decompiler models trained with this new corpus substantially outperform previous baselines on execution correctness and code quality.

**Dataset Code Accessibility:**

Yes

**Dataset Code Comments:**

All code and data are open with detailed documentation.

**Ethical Considerations:**

No, there are no or only very minor ethics concerns

**Final Justification:**

After reading the rebuttal and considering the comments from other reviewers and the authors’ responses, I believe the authors have adequately addressed the concerns raised. Therefore, I have decided to raise my score.

**Limitations Weaknesses:**

1. Although the CTF pipeline does a lot to weed out mismatches, there are probably still some incorrect or noisy binary–source pairs, since function mapping using DWARF and AST heuristics is not always reliable (especially with aggressive optimizations or weird build systems).
2. The focus is only on C/C++; it’s not clear how easily this approach would extend to languages like Rust, Go, or even C++ projects that don’t use CMake.
3. For the metrics (re-executability, readability (R2I), and edit similarity) used in the study, they don’t always capture what reverse engineers or vulnerability researchers actually care about. For example, can the model recover meaningful variable names, complex control flow, or types? Some qualitative analysis or case studies would help here.

**Strengths Contributions:**

1. The manuscript is well-structured.
2. The authors provide the sheer scale and realism of Decompile-Bench. Previous datasets either relied on contest problems or generated toy code, neither of which really reflects the messy reality of real-world binaries. The use of permissively licensed projects, multiple optimization levels, and robust deduplication/cleaning means this data is actually useful and likely to stand the test of time. The technical solution to reliably match binary and source functions—especially considering missing debug info and inlining—seems well thought out, even if it isn’t bulletproof. The leakage-resistant evaluation set (using projects released after 2025 and hand-translated standard benchmarks) is also a nice touch and shows the authors are thinking carefully about avoiding data contamination.

---

> ### Author Rebuttal · Authors · 2025-07-31
>
> Thank you for the encouraging feedback and helpful suggestions. Please find the itemized responses to the questions below.
>
> **Q1:** ...CTF pipeline...are probably still some incorrect or noisy binary–source pairs.
>
> **A1:** Thank you for raising this important concern. To verify the correctness of binary-source matching, we conducted a two-part evaluation: a direct, qualitative analysis and an indirect, empirical validation through downstream task performance.
>
> 1.Direct Qualitative Evaluation:
>
> A key challenge is that a "ground truth" one-to-one mapping often doesn't exist. Therefore, we performed a rigorous qualitative evaluation, which we argue is rather fit and meaningful for this task. We manually inspected a random sample of 1,000 pairs from our dataset. Our analysis confirmed that Algorithm 1 works as intended, with the observed "noise" falling into two distinct and expected categories:
>
> **Negligible Parser Noise (<0.1% of cases):** In extremely rare instances, the source code parser (tree-sitter) makes minor segmentation errors (e.g., with nested functions). These cases are statistically insignificant and unlikely to impact training effectiveness.
>
> **Compiler-Induced Scope Mismatches (~25% of cases):** A far more common scenario arises from aggressive compiler optimizations, particularly function inlining. This results in a binary function that correctly contains code from its primary source function plus code from other inlined functions. This is not an algorithmic error but an authentic and unavoidable artifact of real-world compilation. We consider these pairs to be **valuable and necessary training data** that teaches the model to handle the complexities of optimized binaries.
>
> 2.Indirect Empirical Validation:
>
> The ultimate test of the data's quality is its **effectiveness in training**. Our model, LLM4Decompile-DCBench, which was fine-tuned on the dataset generated by Algorithm 1, achieved a 20% relative performance gain over its base model. Such a significant improvement would be highly unlikely, if the binary-source pairs were noisy or incorrectly matched. This strong downstream result serves as powerful empirical evidence that the pairs generated by our algorithm are of high quality and correctness.
>
> In summary, our claim of precision is backed by both direct inspection and strong empirical results. The algorithm correctly links a binary to its true source origin, and the "noise" it captures is not an error but an essential feature of the problem space that leads to a more robust final model.
>
>
> **Q2:** ...this approach would extend to languages like Rust, Go, or even C++ projects that don’t use CMake.
>
> **A2:** Thank you for your feedback. We designed our pipeline to be modular, which makes our framework highly adaptable.
>
> First, for other compiled languages, the core requirement is to adapt the compilation stage (inherently language dependent) and the DWARF parser for the target language's specific conventions.
>
> Rust: Would require minimal changes, as its DWARF debug format is highly similar to C/C++.
>
> Go: Would require a custom DWARF parser due to its unique format, but our core tracing and alignment algorithms would remain applicable.
>
> We plan to incorporate support for Rust and Go in our future work.
>
> Second, regarding build systems for C/C++, we successfully **added support for projects using Makefiles**, which, along with CMake, represent the vast majority of C/C++ build systems. Limited by the time, we're only able to process around 10% of these makefile tasks, with **83K pairs (condensed from 5M pairs) from 1700 permissive C/C++ projects**, and have made the new data publicly available in our collection on Hugging Face. We will continue to process the data during the revision.
>
> **Q3:** ...can the model recover meaningful variable names, complex control flow, or types? Some qualitative analysis or case studies would help here.
>
> **A3:** This is an excellent point. To address this gap, we have now added both a detailed case study and a new evaluation using GPT-4 as a judge to score the recovery quality.
>
> 1.Manual Case Study Analysis:
>
> We begin with a representative example from our test set to provide a concrete illustration of the differences between decompilers.
>
> a.Source code:
> ```
> my_bool bitmap_is_set_all(const MY_BITMAP *map)
> {
>   my_bitmap_map *data_ptr= map->bitmap;
>   my_bitmap_map *end= map->last_word_ptr;
>   for (; data_ptr < end; data_ptr++)
>     if (*data_ptr != 0xFFFFFFFF)
>       return FALSE;
>   return (*data_ptr | map->last_word_mask) == 0xFFFFFFFF;
> }
> ```
>
> b.Assembly code (input to LLM-based decompilers):
> ```
> bitmap_is_set_all:
> pushq %rbp
> movq %rsp, %rbp
> movq %rdi, -0x10(%rbp)
> ...[30 lines]
> popq %rbp
> retq
> nopl (%rax,%rax)
> ```
>
> c.IDA Pro decompile result
> ```
> bool  bitmap_is_set_all(long long a1)
> {
>   _DWORD *i; // [rsp+8h] [rbp-18h]
>   for ( i = *(_DWORD **)a1; (unsigned long long)i < *(_QWORD *)(a1 + 8); ++i )
>   {
>     if ( *i != -1 )
>       return 0;
>   }
>   return (*(_DWORD *)(a1 + 24) | *i) == -1;
> }
> ```
>
> d.GPT4.1-mini decompile result
> ```
> bool bitmap_is_set_all(bitmap_t* bitmap) {
>     uint64_t* current = bitmap->addr;
>     uint64_t size_in_words = bitmap->size / sizeof(uint64_t);
>     for (uint64_t i = 0; i < size_in_words; ++i) {
>         if (current[i] != UINT64_MAX) {
>             return false;}}
>     return true;
> }
> ```
>
> e.LLM4Decompile-End decompile result:
> ```
> bool bitmap_is_set_all(const struct bitmap *b)
> {
>     const struct bitmap_header *h = (const struct bitmap_header *)b;
>     const unsigned int *p = (const unsigned int *)h;
>     const unsigned int *end = (const unsigned int *)h->end;
>     bool ret;
>     if (p < end) {
>         if (*p != 0xffffffff) {
>             ret = false;
>         } else {
>             p++;
>             while (p < end) {
>                 ret = (*p++ | h->mask) == 0xffffffff;}}
>     } else {
>         ret = true;
>     }
>     return ret;
> }
> ```
>
> f.LLM4Decompile-DCBench decompile result:
> ```
> bool bitmap_is_set_all(const struct bitmap *bitmap)
> {
>     const uint8_t *p = bitmap->data;
>     const uint8_t *end = bitmap->end;
>     while (p < end) {
>         if (*p != UINT8_MAX)
>             return false;
>         p++;
>     }
>     return (*p | bitmap->mask) == UINT8_MAX;
> }
> ```
>
> Our analysis shows:
>
> IDA Pro: While correctly preserving the program's control flow, its output lacks readability. It produces hard-to-read type conversions with pointer arithmetic (e.g., "(_DWORD *)(a1 + 24))" and generic, uninformative variable names.
>
> GPT: This model excels at generating plausible, human-readable variable names and types (e.g., bitmap_t bitmap, which is close to the ground truth MY_BITMAP map). However, it often fails on logical correctness, in this case, hallucinating a "return true;" statement and ignoring the critical return comparison logic.
>
> Our Models (LLM4Decompile-End & -DCBench): Both models successfully balance these two aspects. They recover the correct control flow and logic while simultaneously generating meaningful variable names and types. Our final model, LLM4Decompile-DCBench, demonstrates superior insight, for instance, by naming a variable const uint8_t *end = bitmap->end, which accurately reflects its functional role.
>
> 2.Automated Qualitative Analysis:
>
> To validate these findings systematically across our entire Github2025 dataset, we introduced a typical evaluation methodology. We utilized GPT-4 as an expert judge to score the output from each decompiler on a scale of 1 (poor) to 100 (excellent) across three qualitative areas: variable name recovery, control flow clarity, and type reconstruction (Due to space limitation, the prompt for the GPT evaluation has been uploaded to the anonymous GitHub page). This approach allows us to quantify these subjective aspects at scale.
>
> | Variable naming       | O0    | O1    | O2    | O3    | Average |
> |-----------------------|-------|-------|-------|-------|---------|
> | GPT-4.1-mini          | 48.99 | 42.24 | 43.07 | 39.98 | 43.57   |
> | IDA                   | 33.66 | 27.16 | 29.49 | 28.99 | 29.83   |
> | LLM4Decompile-End     | 64.15 | 63.48 | 62.39 | 63.84 | 63.47   |
> | LLM4Decompile-DCBench | 76.38 | 77.18 | 77.53 | 76.69 | 76.95   |
>
> | Control flow          | O0    | O1    | O2    | O3    | Average |
> |-----------------------|-------|-------|-------|-------|---------|
> | GPT-4.1-mini          | 63.25 | 50.09 | 50.41 | 50.10 | 53.46   |
> | IDA                   | 63.28 | 59.42 | 60.35 | 60.62 | 60.92   |
> | LLM4Decompile-End     | 73.75 | 73.49 | 73.61 | 74.65 | 73.88   |
> | LLM4Decompile-DCBench | 83.61 | 85.13 | 85.56 | 84.87 | 84.79   |
>
> | Type recovery         | O0    | O1    | O2    | O3    | Average |
> |-----------------------|-------|-------|-------|-------|---------|
> | GPT-4.1-mini          | 55.69 | 45.18 | 46.75 | 44.93 | 48.14   |
> | IDA                   | 63.53 | 60.37 | 62.29 | 61.67 | 61.97   |
> | LLM4Decompile-End     | 76.47 | 77.82 | 78.98 | 77.42 | 77.67   |
> | LLM4Decompile-DCBench | 80.13 | 82.26 | 82.22 | 81.55 | 81.54   |
>
> The results, presented in the updated results table, strongly support our case study findings:
>
> IDA struggles with semantic richness, achieving a variable naming score that is only 68% of our model's.
>
> GPT fails on logical integrity, on control flow recovery, it scores only 63% of our model's.
>
> Our fine-tuned models significantly outperform all baselines. Notably, LLM4Decompile-DCBench improves upon the strong LLM4Decompile-End base model by 21.2% in name recovery, 14.7% in control flow, and 5.0% in type reconstruction.
>
> In summary, this comprehensive qualitative analysis demonstrates that while traditional tools preserve logic at the cost of readability, and general LLMs sacrifice logic for readability, our approach successfully achieves both. This makes our model's output significantly more valuable for the practical tasks that reverse engineers and vulnerability researchers care about.

---

> > ### Comment · Reviewer_xHmv · 2025-08-05
> >
> > Thank you for the detailed response. I believe the authors have addressed my concerns, and I have decided to increase my score.

---

> > > ### Author Response · Authors · 2025-08-05
> > >
> > > Thank you for your valuable time, we are glad to hear that our responses have addressed your concerns and are grateful for the updated score!
> > >
> > > We will continue to process the data and release it as soon as it's ready.

---

### Note · Authors · 2025-08-12

We sincerely thank the area chair and reviewers for their valuable time and effort. We are grateful for the positive and encouraging feedback from the reviewers.

We are pleased that our efforts to build a **large-scale binary-source benchmark were widely appreciated**, with reviewers highlighting it as a "valuable resource" (reviewers y9L1, X9M1), "useful and likely to stand the test of time" (reviewer xHmv), and of "interest to a wider range of researchers" (reviewer jJhk). The rigor of our **extensive experiments and multiple evaluation metrics was also recognized** (reviewers y9L1, jJhk), along with our work on a **leakage-resistant evaluation set**, which was noted as "a nice touch" (reviewer xHmv).

During the rebuttal period, we addressed all reviewer concerns, and **reviewers confirmed that their issues had been resolved**. Furthermore, the ethics reviewers have concluded that **our paper presents no ethical concerns**. The key enhancements to our work are as follows:

**Dataset Expansion:** We released 643K new data pairs (condensed from 12M pairs) from 2,600 permissive C/C++ projects.

**Algorithm Correctness:** We verified the correctness of our binary-source matching algorithm, demonstrating that it correctly links binaries to their true source origins. The "noise" it captures is not an error but rather an essential feature of the decompilation.

**Qualitative Analysis:** We added a detailed case study and a new evaluation using GPT-4 to score recovery quality. This study demonstrates our approach's unique ability to preserve both logical integrity and readability, a balance not achieved by traditional tools or general LLMs.

**Re-executability Analysis:** We provide a constructive analysis on creating executable GitHub test cases. Our findings offer the community a clear perspective on two key factors for building robust, leak-free benchmarks: the need to address pervasive data leakage in existing test suites and the high cost required for manual construction.

**Additional Experiments:** We have incorporated two key baselines (Claude-Sonnet-4-reasoning and Idioms-1.3B) and evaluated all models on an independent benchmark to provide a more comprehensive comparison.

We again thank our reviewers for their insightful feedback, which has significantly improved our work.

---

### Decision · Program_Chairs · 2025-09-18

**Decision:**

Accept (poster)

**Comment:**

The work introduces Decompile-Bench and Decompile-Bench-Evals to accelerate the progress of LLM-based decompilers. Reviewers appreciate the size (2M binary source function pairs) and practicalness (from real-world permissively licensed GitHub repos). The work has done quite thorough analysis (algorithm correctness, qualitative analysis, etc) and also demonstrates that by fine-tuning on the dataset, 20% improvement can be achieved in terms of the re-executability rate, which is impressive. Ethical issues have also been properly addressed.

===== FINAL UPDATE FROM DB Track PCs ====

The final decision for this paper has been taken by the program chairs after consultation with the SACs. All Senior Area Chairs have ranked papers according to the feedback from the AC during the review process. We decided to leave the original meta-review to reflect the opinion of the AC in light of the initial discussions with reviewers and SAC.